# Impact-based forecasting of tropical cyclone-related human displacement to support anticipatory action

Pui Man Kam [1,2] ✉, Fabio Ciccone[1], Chahan M. Kropf [1,3], Lukas Riedel [1,3], Christopher Fairless[1] & David N. Bresch [1,3]

Tropical cyclones (TCs) displace millions every year. While TCs pose hardships and threaten lives, their negative impacts can be reduced by anticipatory actions like evacuation and humanitarian aid coordination. In addition to weather forecasts, impact forecast enables more effective response by providing richer information on the numbers and locations of people at risk of displacement. We introduce a fully open-source implementation of a globally consistent and regionally calibrated TC-related displacement forecast at low computational costs, combining meteorological forecast with population exposure and respective vulnerability. We present a case study of TC Yasa which hit Fiji in December 2020. We emphasise the importance of considering the uncertainties associated with hazard, exposure, and vulnerability in a global uncertainty analysis, which reveals a considerable spread of possible outcomes. Additionally, we perform a sensitivity analysis on all recorded TC displacement events from 2017 to 2020 to understand how the forecast outcomes depend on these uncertain inputs. Our findings suggest that for longer forecast lead times, decision-making should focus more on meteorological uncertainty, while greater emphasis should be placed on the vulnerability of the local community shortly before TC landfall. Our open-source codes and implementations are readily transferable to other users, hazards, and impact types.

Human displacement occurs when people are forced to leave their homes or places of habitual residence due to external events such as natural extreme weather hazards[1]. Every year, weather extremes cause millions of people around the world to be displaced. Tropical cyclones (TCs) account for the second largest contribution to these human displacements after floods, with an average of 9.3 million people being displaced every year between 2017 and 2020[2]. The duration of displacement and its humanitarian impact vary widely: from short-term pre-emptive evacuations to long-term displacement if houses or community infrastructures are significantly damaged, or people lose access to their economic activities. To cope with the post-disaster recovery, international and national assistance for humanitarian response and relief funds are often required. The range of assistance includes providing emergency shelter, clean water and food, health care, psychological support, and long-term community and livelihood recovery and restoration[3,4].

Anticipatory actions can help to reduce the negative impacts of extreme weather events. Examples of anticipatory actions include evacuation planning, emergency protection, and humanitarian aid coordination. The World Meteorological Organization (WMO) proposed the WMO Coordination Mechanism (WCM)[5] and a pilot project Weather4UN[5] that enable access to weather and climate information and

[1]Institute for Environmental Decisions, ETH Zürich, Zurich, Switzerland. [2]Internal Displacement Monitoring Centre, Geneva, Switzerland. [3]Federal Office of Meteorology and Climatology MeteoSwiss, Zurich, Switzerland. ✉e-mail: mannie.kam@usys.ethz.ch

the provision of expert advice from WMO Members to the member states and other humanitarian agencies. Another similar scheme is the Forecast-based financing (FbF) proposed by the International Federation of Red Cross and Red Crescent Societies (IFRC)[6]. It is a specific financial scheme that allows access to financial resources based on scientific forecasts and risk analysis for humanitarian actions agreed in advance. The allocation of the funds is automatically released when a certain forecast threshold is reached, which allows mitigation measures to take place prior to the hazard and thus reducing impacts[7].

Currently, the planning for anticipatory action is mostly based on weather forecasts, which may potentially miss crucial nuances required to minimise negative impacts. For instance, knowing whether the same intensity of storm hits a populated city or a scattered rural area, or being aware of the resilience of the directly affected community, will in general lead to different preemptive measures. At the moment, this auxiliary information is indirectly taken into account by decision-makers who use expert knowledge and past experiences to guide their assessments, but there is rarely a systematic quantification available.

An impact forecast moves a step forward from conventional weather forecasts to give information on how the weather will affect people. It systematically translates the weather information into risk by combining it with social variables[8]. There have been multiple emergency evacuation decisions support tools for tropical cyclones in the US that combine weather forecast with traffic information to support evacuation plannings (e.g. Harris et al.[9], Davidson et al.[10], and Blanton et al.[11]), and establishing platforms that provides weather risk communication with integrates societal information flow (e.g. CHIME; Morss et al.[12]). Currently, there are however no tools that provide globally consistent impact information for human displacement. Here we introduce a proof-of-concept implementation of a global TC-related displacement impact forecast, that could provide more comparable, standardised, and less singular information to support decision-making for anticipatory action.

We use the open-source probabilistic natural catastrophe risk assessment platform CLIMADA (CLIMate ADAptation)[13] for the displacement impact forecast. The platform is widely used in long-term weather and climate impact assessments for building insurance and other socioeconomic impacts[14–20], and has also been used for establishing operational impact forecasts for building damages from winter storms[21]. CLIMADA generates risk information and quantifies socio-economic impacts by integrating hazard, exposure, and vulnerability data. In our study, we forecast the number of people who are at risk of being displaced due to upcoming TC events by combining datasets from i) TC track forecasts and their associated windfields as hazard, ii) the global population distribution as exposures, and iii) the vulnerability of people being displaced represented by a univariate impact function relating the TC wind speed to the probability of displacement at a given location, see Fig. 1.

In typical disaster risk assessments, risks are quantified by considering long-term aggregated average impacts from past events[13]. However, when it comes to forecasting a single event, we argue that it is important to not only consider the most probable outcome but also look at the overall distribution of plausible forecast outcomes when deciding on the anticipatory action. The uncertainties in the forecasted impact arise from the complex interplay between the meteorological forecast variability based on the ensemble forecast system[22,23], and uncertainties associated with all other input data. We will show that it is crucial to account for uncertainties in all inputs to provide a more comprehensive perspective on the forecasted outcomes. Furthermore, to make this uncertainty practical in the context of decision-making, we conduct sensitivity analyses to gain a deeper understanding of how the relevant outcomes depend on these uncertain inputs.

Here we demonstrate the TC impact forecast for displacement by performing an analysis for the TC Yasa that hit Fiji in December 2020 and caused the displacement of 23,414 people[2,24]. We chose Fiji as a demonstrating case as the Pacific Islands are an under-studied region, and while the island characteristics enable a showcase of hit-or-miss scenarios, the probability of whether there will be impacts are important when providing the forecast information.

We first forecast risk of displacement by considering only the meteorological forecast variability. Then we quantify the full uncertainty range of forecast outcomes that includes uncertainty from the exposures and vulnerability. We repeat the calculation using a quasi-Monte Carlo sampling method to cover a wide range of possible input variations[25,26], and a sensitivity analysis to attribute which input variations contribute most to the overall uncertainty. We also generalise the main findings by performing the same displacement forecast analysis at different lead times prior to the landfall for all past TC events which have displacement records from 2017 to 2020[2]. Our results provide decision-makers with a wide range of possible outcomes and guidance on which input variations should be considered for better-informed decisions on anticipatory actions.

## Results
### Impact forecast for TC Yasa
Here we first demonstrate the displacement forecast by performing a case study of the TC Yasa, which passed through Fiji on 17 December 2020. TC Yasa reached category 5 of the Saffir-Simpson wind scale, with maximum sustained winds reaching 146 kts (75.1 m/s) before making its landfall in Fiji[27]. The cyclone was the strongest since TC Winston in 2016 and was the most destructive cyclone in Fiji during the cyclone season 2020–2021, damaging infrastructure, buildings and agricultural areas[24]. IDMC has recorded 23,414 people on the Fijian islands as being displaced[2].

Figure 2a shows a spatially explicit map of the forecast-ensemble-averaged displacement in Fiji, overlaid with the 51 ensemble forecast TC tracks from the Integrated Forecasting System maintained by the European Centre for Medium-Range Weather Forecasts (ECMWF-IFS)[22,23]. The exposure is the static estimation of population distribution as described in subsection "Population exposures" of section "Methods", and the impact function (or vulnerability curve) is a calibrated function that minimises the relative errors over all the recorded displacement events in the Pacific Island region, detailed in subsection "Calibration of impact functions for human displacement" of section "Methods". The average forecasted number of people at risk of displacement in Fiji is 172,000 (orange dashed line in Fig. 2b), whilst the total number of displaced people ranges from 3500 to 450,000 based on the maximum 1-minute sustained wind fields calculated from the 51 ensemble forecast TC tracks. The distribution of the number of displaced people per ensemble member, as shown in Fig. 2b, is right skewed with a long tail of high impacts. The large displacement uncertainty here only comes from the spread of the TC tracks. The magnitude of the impact mainly depends on whether the TC hits the main island and passes through the populated cities, or remains mostly over the ocean. The case that gives the highest number of people who could be potentially displaced features a forecasted TC track that crosses Fiji's main island of Viti Levu near the populated cities such as Suva, Lautoka and Nadi, whereas the lowest numbers come from tracks that do not make landfall and pass the Northeastern side of Fiji.

### Global uncertainty and sensitivity analysis
An effective impact forecast should not only include the variability of the meteorological forecast, but also the uncertainties within the exposure information and impact functions[21,25,28]. Here we include the uncertainties from the total population count and replace the single impact function with an ensemble of functions, with each member calibrated against a past event to represent the range of plausible functions (detailed in subsections "Calibration of impact functions for human displacement" and "Uncertainty and sensitivity" of section "Methods"). Note that this calibration method differs from the one

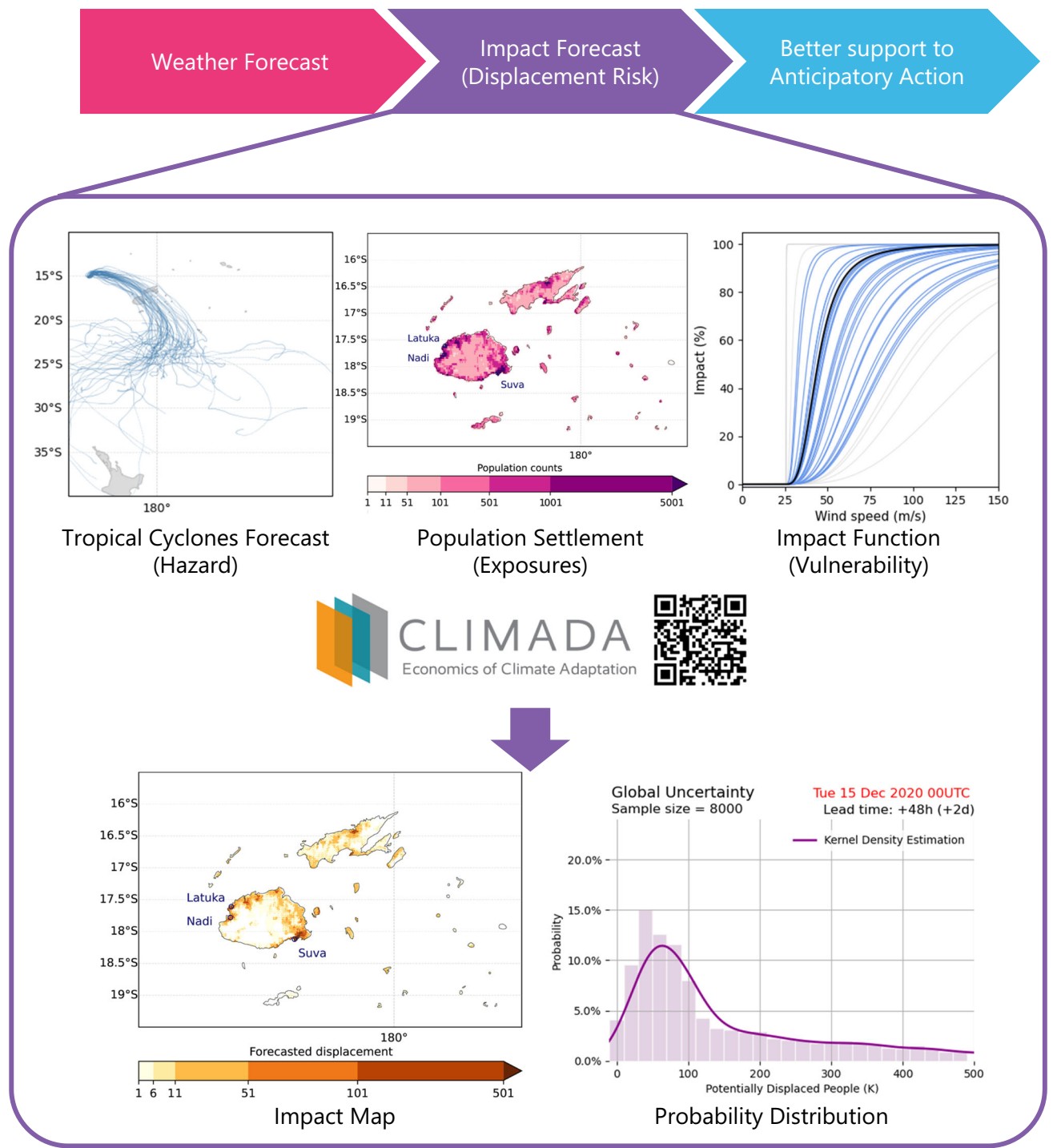

**Fig. 1 | Schematic illustration of the displacement impact forecasting system implemented in CLIMADA.** The spatially explicit risk of displacement is calculated based on weather forecasts as hazard, population settlement as exposures, and vulnerability information. CLIMADA facilitates the production of different summarising risk metrics and plots[13,25]. Maps and plots are plotted using python package Cartopy and Matplotlib[55,56].

used for Fig. 2, as discussed in section "Discussion". We ran the impact forecast model more than 8000 times with each run randomly sampling parameters for each model component (meteorological forecast, exposures, and impact functions). The computational time of this uncertainty distribution starting from the TC forecast tracks extraction is around 10 minutes on a MacBook Pro equipped with 2.3 GHz Quad-Core Intel Core i7 and 16 GB random-access memory (RAM).

Figure 3a shows the probability distribution of the global uncertainty analysis. The resulting distribution is even more right-skewed compared to 2b) (meteorological variability plot), yielding an average

of 123,391 people in Fiji being displaced by TC Yasa, but with the average and peak probability closer to the reported number of displacements by the IDMC.

Furthermore, the corresponding first-order sensitivity analysis shown in Fig. 3 indicates that the overall impact forecast uncertainty is most sensitive to the uncertainty from impact functions (sensitivity index 0.411), with the meteorological forecast uncertainty also making a substantial contribution (0.320) at two days lead time of the forecast. The sensitivity index for the total population, on the other hand, is very small (0.005).

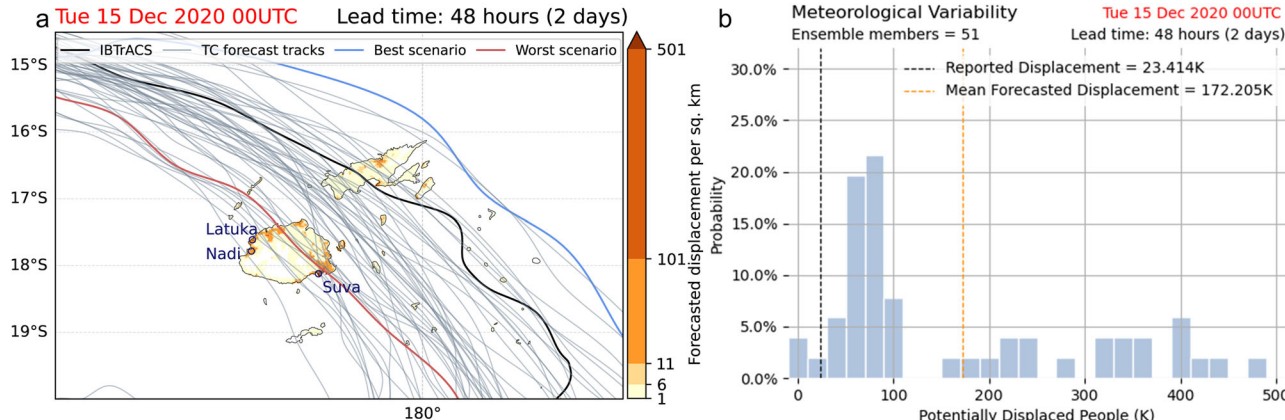

**Fig. 2 | Forecasted displacement in Fiji by tropical cyclone Yasa. a** The forecast-ensemble-averaged impact map[55,56] of displacement by TC Yasa in Fiji as forecasted at 00:00 UTC on 15 December 2020, 2 days before the TC landfall. The black line shows the observed best track of TC Yasa from IBTrACS[27]. Grey lines show the ensemble of ECMWF forecasted TC tracks, with the blue and red lines indicating the best and worst case scenarios with respect to the forecasted total number of displacements. **b** Distribution of the forecasted potential number of displacements in Fiji based on wind fields calculated from 51 ensemble member tracks shown in (**a**)[56]. The vertical lines indicate the reported (black) and the mean forecasted (orange) displacement.

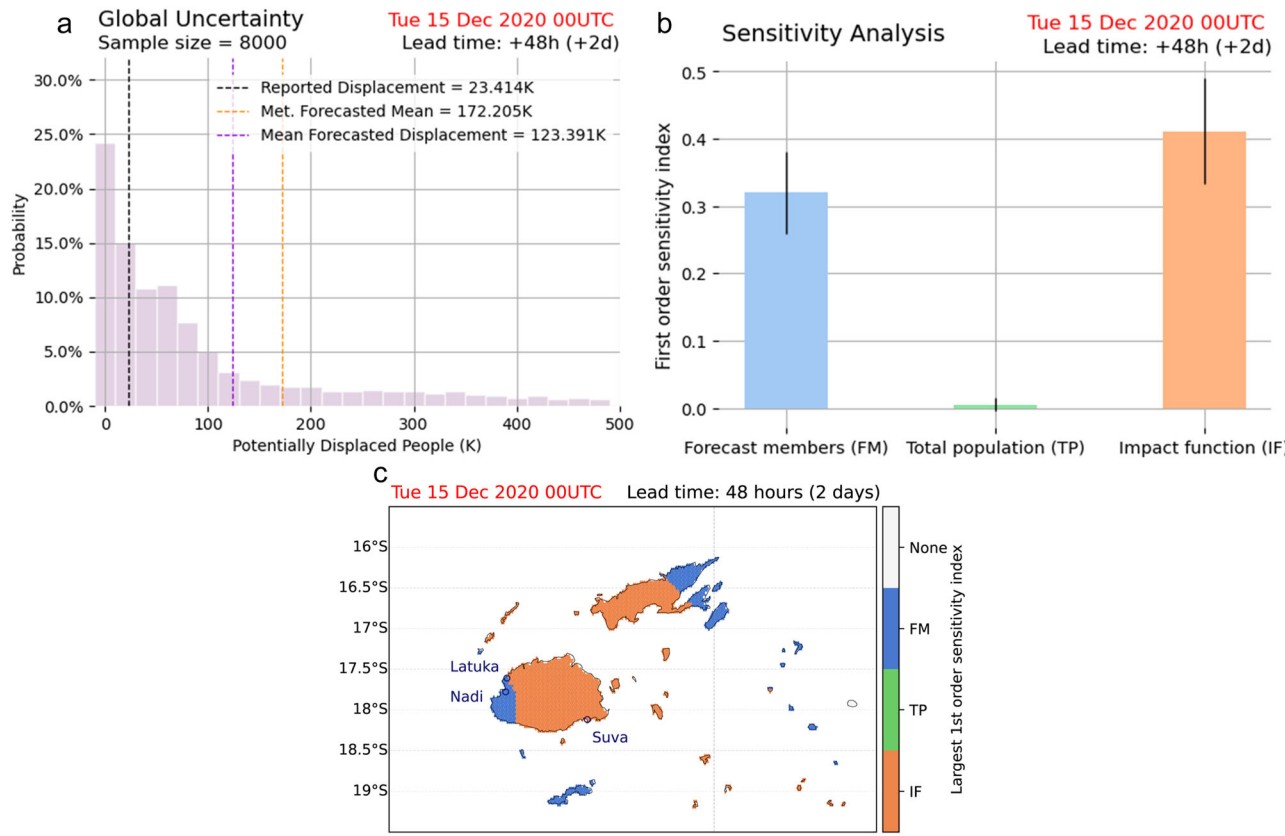

**Fig. 3 | Uncertainty and sensitivity analysis for the forecasted TC displacement in Fiji. a** Probability distribution of the forecasted potential number of displaced people in Fiji due to TC Yasa with two days' lead time for each impact model run in the global (including exposure, hazard and vulnerability uncertainty) uncertainty and sensitivity analysis. The black dashed line indicates the number of reported displacements from IDMC. The orange dashed line represents the forecasted mean from the 51 ensemble members of the TC forecast (from Fig. 2a), and the purple dashed line represents the mean forecasted displacement from the global uncertainty and sensitivity analysis[56]. **b** The Sobol first-order sensitivity indices for the total number of displaced people, the error bars represent the 95th percentile confidence interval for each index obtained from the Saltelli[57] algorithm[56]. **c** The largest Sobol first-order sensitivity indices at each grid point[55–57].

## Displacement impact forecast at different lead time

We repeat the preceding analysis for lead times ranging from 3.5 days until just before landfall at intervals of 0.5 days. Figure 4a shows the kernel density fit to the probability distribution of the global uncertainty of the displacement impact forecast at different lead times. The distribution is more spread out for longer lead times, but narrows for the forecast close before landfall.

Figure 4b shows the first-order Sobel sensitivity index[29] of all the uncertain input parameters. We observe a general decreasing trend of the sensitivity index for the meteorological forecast, while an

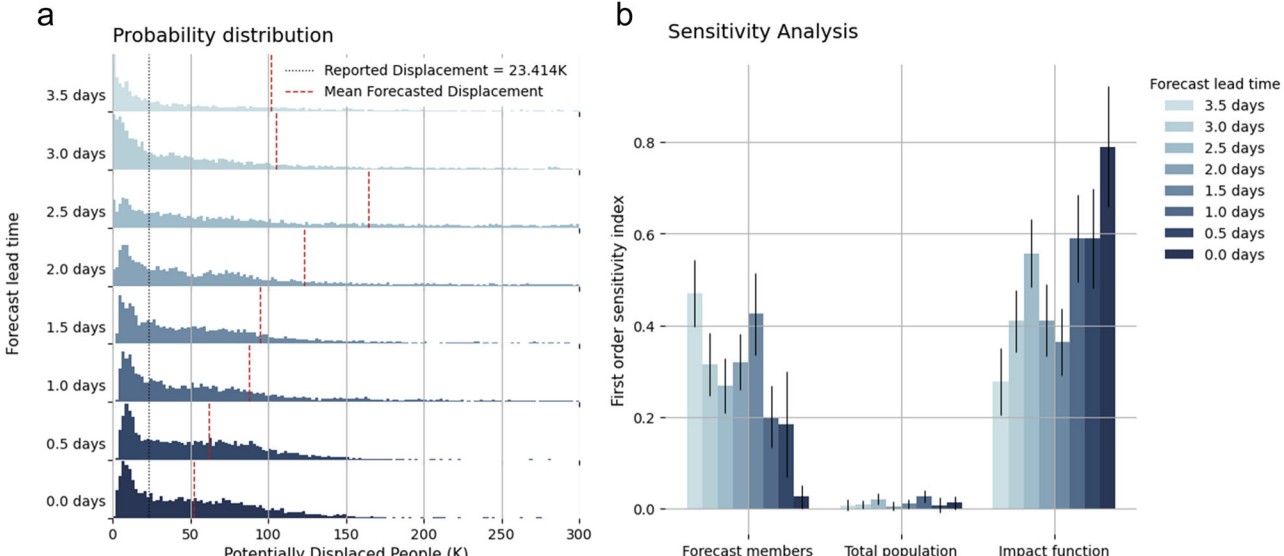

**Fig. 4 | Uncertainty and sensitivity analysis for the forecasted TC displacement in Fiji at different forecast lead time. a** Probability distribution of the impact forecast at different forecast lead times ranging from 3.5 days to 0 days from the landfall of TC Yasa at Fiji, with the black dashed line indicating the number of reported displacement from IDMC and the red dashed line the mean forecasted displacement[56]. **b** First-order sensitivity indices of the different uncertainty parameters for the total number of displaced people at different forecast lead times. The error bars represent the 95th percentile confidence interval for each index[56].

increasing trend for the impact function. In the meantime, the index for the change in total population remains close to zero at all forecast lead time. We remark that the increase in sensitivity to the vulnerability uncertainty at shorter lead times cannot be equated with an increase in uncertainty from vulnerability, but only that the relative contribution to the total uncertainty from vulnerability increases, and from forecast decreases.

## Uncertainty and sensitivity analysis for past TC displacement events from 2017 to 2020

We ran the same uncertainty and sensitivity analysis on the total number of displaced people for worldwide recorded TC events from 2017 to 2020 at lead times from 3 to 0.5 days before landfall. We find the impact forecast for displacement show a bias of overestimation when comparing to the reported number of displacement from IDMC (see Fig. S6.) Details of the impact forecast validation and biases are included in the supplementary information S4.

Figure 5 summarises the distributions of first-order sensitivity indices for all TC events. Similar to the trends identified in Fig. 4b, the median of the first-order sensitivity index for meteorological forecast is the largest when the forecast lead time is longer, but decreases closer to the TC landfall. The impact function's sensitivity index, in contrast, is smaller with longer lead times but increases over time. Hence, on average over all considered events, we find that the relative contribution to the overall uncertainty from the forecast is largest at long lead times, and from vulnerability at short lead times. This overall trend does however not hold for all individual cases as exemplified with TC Harold in the Supplementary Figs. S4 and S5 for which also at long lead times the sensitivity to the vulnerability uncertainty is largest. Note that the forecast uncertainty for TC Harold is comparable to TC Yasa even though the sensitivities are different.

## Discussion

Our work shows that it is possible to provide spatially explicit impact forecasts for tropical cyclones-related displacement from publicly available data in near-real time. We combine the tropical cyclone (TC) track forecasts from ECMWF with the population exposure and vulnerability of displacement in the open-source probabilistic risk assessment platform CLIMADA to forecast the displacement risk by TCs. We demonstrate the impact forecast for displacement with TC Yasa that hits Fiji in December 2020. We argue that our probabilistic approach is more useful than a deterministic model. The spread of the displacement predictions reflects the statistical uncertainty in the model, and it is important to consider the full uncertainty distribution when making decisions for anticipatory action, rather than only employing the often-used ensemble-averaged metrics. We show that the modelled impact follows a strongly skewed distribution, and thus the averaged impact cannot reflect the full range of possible outcomes. Importantly, the tail risk of high-impact scenarios, which can be the triggers for major disasters, is inadequately represented by average measures and must be derived from a quantified uncertainty distribution.

Uncertainty around the impact forecast stems from the interplay between meteorological forecast variability, uncertainties from the population number estimation, and the spread of the impact functions. In general, it is difficult to exhaustively characterise all uncertainties of input data[25,28], and one should always carefully consider which elements are needed for the purpose of the modelling task[30]. For impact-based forecast for anticipatory action planning, it is important to understand what individual events are plausible (e.g., will the TC hit the Eastern or the Western part of the area?) instead of summary values (e.g., the median outcome of the TC forecast impacts is in both the Eastern and Western areas), where the latter is more common in risk assessment. The chosen uncertainties in this study reflect this purpose where data is available. For the hazard, we consider the individual outcomes from each forecast ensemble member as a representation of the meteorological uncertainty.

For the vulnerability, qualitative impact data are often not available. To the best of our knowledge, IDMC provides the most comprehensive and consistent source of global displacement data, and hence we use the IDMC displacement data for impact function calibration for the global consistency. At the same time, despite many efforts from IDMC dedicated to systematically collecting and harmonising displacement data, data gaps and inconsistencies remain[31]. Some of these gaps stem from the variability in reporting, for instance, displacement can be described as "homeless" or "moved", or included in the category "directly affected"[31]. In order to capture all the

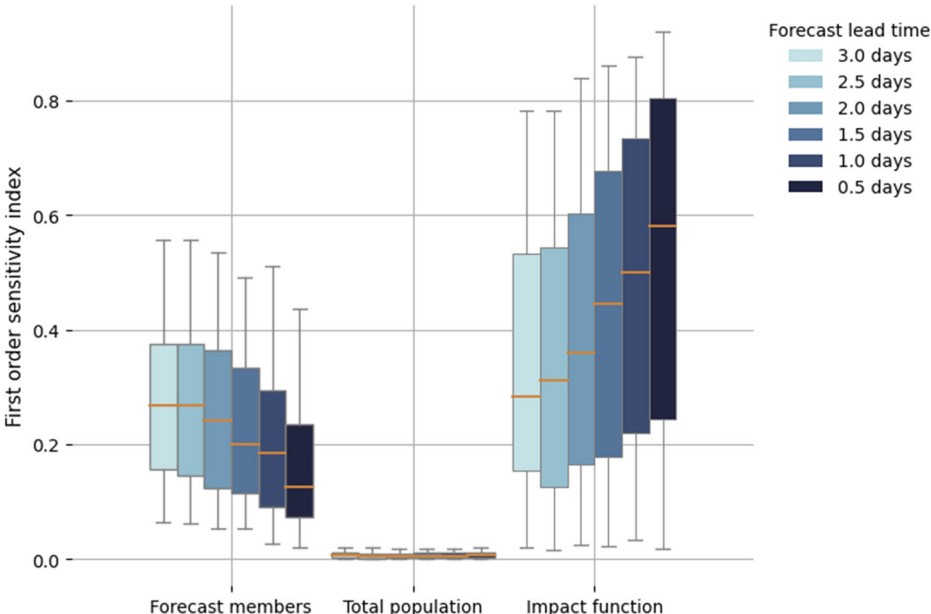

**Fig. 5 | Sensitivity analysis of the TC displacement forecast for events between 2017–2020.** First-order sensitivity indices at different lead time from +3.0 days to +0.5 days for all tropical cyclone events causing displacement between 2017 and 2020 (174 total number of events). The orange lines represent the median of the indices, the boxes show the inter-quartile range, and the whiskers show the 95th percentile of the distribution[56].

plausible displacement outcomes from the TC events, we consider the ensemble of individual impact functions calibrated to best represent each recorded displacement event in the used IDMC database (per region). This is in contrast to calibrating one optimal impact function that minimises errors for all past events at once (which is the impact function used in estimating displacement with meteorological uncertainty only in subsection "Impact forecast for TC Yasa" of section "Results" and Fig. 2), and subsequently sample the uncertainty from the confidence interval of the optimisation process itself. The latter is typically done in climate risk assessment studies and is best suited to estimate the average impact over many events and its uncertainty.

For the exposures, due to a lack of uncertainty information from the population data layer provider, we only vary homogeneously the total population, which can be understood as a proxy for the movement of people in the area. Importantly, our analysis demonstrates the importance of performing a global uncertainty analysis (i.e., varying all input data simultaneously)[28] since it is the interplay of hazard, exposure and vulnerability that can lead to extreme scenarios (e.g., a strong TC hitting directly highly populated cities with large displacement vulnerability).

To understand how the relevant outputs depend on uncertain inputs we also perform a sensitivity analysis. Conducted at various lead times, we observe that in general meteorological uncertainties play a dominant role at larger forecast lead times, whereas local vulnerability becomes more significant as TCs approach landfall and forecast uncertainty reduces. This finding offers valuable insights for decision-makers, guiding their considerations when devising anticipatory actions at different points in time. In situations with longer lead times, decision-makers may find it beneficial to seek expert input from meteorologists regarding the TC's expected path and further cyclo-genesis. On the other hand, as the TC approaches landfall, involving individuals with local community knowledge becomes crucial for effective planning of anticipatory action. However, we also point out that there are large variations of the sensitivity indices for both meteorological forecast and impact function (error bars in Fig. 5). There can be large lead times where the uncertainty contribution from the meteorological forecast is little but the contribution from the uncertainty of impact functions is large, and vice versa for shorter lead

times. For example, this was the case with TC Harold, which impacted Vanuatu and displaced 80,000 people in April 2020 (further details in Supplementary Information S3). Such instances could be related to the fact that some meteorological situations are easier to predict than others.

Several sources of uncertainty are not represented in our model setup which if included could add more nuances to the previous conclusions from the uncertainty and sensitivity analysis at different lead times. Here we model population displacement as a direct function of TC wind speeds, but we acknowledge that the relationship between displacements and TC hazards is much more complex. On the physical hazard side, displacement occurs not only due to damages from wind, but also due to storm surge, flood and torrential rain. However, since both surges and rain correlate to the TC wind speed[32], the latter is often taken as a proxy of the overall TC hazard intensity[33,34]. In addition, our model does not resolve the compounding risks if multiple hazards occur at the same time or in close succession. We choose the simpler single-hazard approach because TC wind speed can be provided by computationally inexpensive models, and considering multiple and compound hazards strongly increases model complexity[35]. In estimating population distribution, it is crucial to recognise that locational uncertainty stems from the accuracy of estimating densely populated clusters, which could be a significant parameter in uncertainty analysis. However, due to the lack of uncertainty information, we only apply a uniform scaling uncertainty of the total population. This likely explains the low sensitivity to the exposures. It is imperative to understand that this does not mean that the exposures' uncertainties are not important. It only means that the impact forecast is not sensitive to homogeneous scaling of the population. The uncertainty and sensitivity analysis can only represent what is used as input for the model[28].

Furthermore, our model only considers displacement as a direct impact of hazards. But people can also be displaced by indirect impacts such as the loss of access to basic services (e.g., water provision) due to cascading failures of critical infrastructures[18]. Our impact functions are calibrated using the IDMC recorded displacement attributed to TC events which include people who have been displaced due to indirect impacts. Maximum sustained wind speed is generally

considered a good proxy for compound damages afflicted by TCs[32,34]. Thus, the modelled total impact in the area affected by TC wind should reflect displacement from direct and indirect causes, but only in areas of high wind speeds. The model could further be improved by incorporating TC sub-hazard footprints like rainfall and storm surge, or explicitly model displacement triggered by housing damages, loss of livelihoods or loss of access to critical infrastructures[36]. The modelling of the impacts of compound hazards is an ongoing scientific effort (e.g. Rossi et al.[36] and Stalhandske et al.[37]).

We further remark that for the planning of anticipatory action, not only the displaced people, but also the people that may decide to remain in the impacted area[38], or even become trapped there if escape routes are blocked[39,40] may be of relevance. Those trapped are often the most vulnerable, but this effect is not captured in our analysis. Additionally, displacement is only one of the numerous socio-economic impacts related to TCs. While forecasts of asset damages and other impacts are possible with approaches similar to ours, we focus on displacement because it is arguably the most relevant impact for humanitarian agencies. Future research is encouraged to disentangle the complex nature of disaster displacement and incorporate them in the next iterations of impact forecast systems, and to integrate additional types of impact for a more complete picture on impending disasters.

Impact information provided on top of the weather forecasts can support humanitarian communities to prepare properly before hazards strike and to act more efficiently. Our presented disaster impact forecast system is open-source and based on solely open-access data which allows any actor even with limited resources to run the model at a low operational cost. Our code and implementation are also transferable to other users (e.g. meteorological services, businesses, governments, individuals), hazards (e.g., floods, heatwaves, storm surges, drought), and impact types (e.g. displacement, mortality). We hope the versatility of our work encourages the ongoing development of impact forecasting systems and fosters collaborative efforts between the risk modelling community and relevant stakeholders.

## Methods
### Risk assessment platform CLIMADA
We use the open-source probabilistic risk assessment platform CLIMADA v3.3 for our displacement impact forecast written in Python[13,41,42]. CLIMADA is designed to simulate the interactions of weather- and climate-related hazards, the exposure of people or assets to the hazard, and the vulnerability of those exposed people or assets in a globally consistent fashion.

We use the tropical cyclone weather forecast as hazard with the population as exposure and their displacement vulnerability as an impact function to predict the number of people at risk of displacement. We forecast the spatially explicit displacement number at 150 arc second resolution on land (~4 km) globally.

### TC track ensemble forecasts and derivation of TC windspeed as hazard
We take the maximum 1-minute sustained wind speed at 10 m above the ground as a proxy of the TC hazard intensity. The TC wind speed is derived from the TC track forecast from the European Centre for Medium-Range Weather Forecasts Integrated Forecasting System (ECMWF-IFS)[22,23], updated in real-time at 00:00 and 12:00 UTC every day. The ECMWF-IFS forecast products consist of a high-resolution deterministic run and 51 ensemble members. We consider only the 51 ensemble members for the displacement risk impact forecast. Each TC track forecast consists of the forecasted positions, central pressure, ambient pressure, and maximum wind speed, available at 6-hour intervals for 240 h, which we further interpolate to 1-hour intervals. The real-time TC forecast tracks are openly accessible from the

ECMWF servers via file transfer protocol (FTP)[43], and are piped to CLIMADA through its tropical cyclone forecast modules. Past TC track forecasts from 2017 to 2020 are available from the THORPEX Interactive Grand Global Ensemble (TIGGE) project[44]. In total, there are 161 displacement events around the globe from the IDMC database with matched ECMWF TC archived forecast tracks[2].

The TC wind speed is then computed at a horizontal resolution of 150 arc seconds on land and 1800 arc seconds on sea from the TC ensemble forecast tracks, based on the revised hurricane pressure–wind model by refs. [13,45].

### Population exposure
For the exposure dataset, we take the spatially explicit representation of the world's population from the Gridded Population of the World (GPW v4) dataset[46], openly accessible from the Socioeconomic Data and Application Center (SEDAC). The dataset is created by uniformly distributing the numbers of people from census data or population figures provided by national statistics offices at the smallest administrative unit, without considering any ancillary sources.

### Calibration of impact functions for human displacement
The vulnerability of people to displacement by TCs is represented by mathematical functions (named impact functions in the CLIMADA terminology, and often called vulnerability curves) relating the TC wind speed to the percentage of people displaced at a location. Prior studies have focused on defining impact functions for TC-related building asset damages in terms of monetary values[33,47,48]. To the best of our knowledge, there is no research yet on impact functions for displacement due to TCs. In general, the impact functions must always be calibrated for the data (exposure and hazard) at hand. This can be done via numerical optimisation and may be complemented by empirical knowledge[49] or expert knowledge[14]. Of central importance is choosing a calibration method suitable for the purpose of the model. Here we assume a sigmoid curve functional form and calibrate a set of impact functions using the reported displacement data from the Internal Displacement Monitoring Centre (IDMC) database (accessed in September 2022). This approach differs from previous approaches that used housing damage as a proxy to estimate displacement[50] and which have been suspected to provide a conservative estimate likely to underestimate displacement[3]. Our approach in particular strives to reproduce the total reported displacement per TC event. It thus implicitly includes the effects of all TC sub-hazards (wind, flood, surge), preemptive measures such as evacuation, as well as regional and cultural differences.

We calibrate the displacement impact functions to the reported displacement data from the IDMC database for events recorded from 2008 to 2020. These reported numbers of displacements can vary from a few persons to millions in some high-impact events[2]. IDMC systematically collects displacement data from governmental and non-governmental institutions post-disasters, and dedicated efforts to verify and harmonise data to ensure their interoperability. All the data are curated in the global database aggregated at country level[2], and the analyses are presented in the IDMC yearly Global Report on Internal Displacement (GRID) (e.g., GRID 2023[51]). In this study, we take the 394 events for which IDMC has reported new displacement due to TCs as the impact data for the impact function calibration.

The hazard events are represented by the 1-min maximum sustained wind speed at 10 m from surface derived from the corresponding historical TC tracks from the International Best Track Archive for Climate Stewardship (IBTrACS)[27], and the associated wind speed computed from the[45] model as implemented in CLIMADA[42]. We note that there might exist biases in terms of the TC hazard intensity between these historical tracks and the ECMWF forecast tracks[52,53] (c.f., subsection "TC track ensemble forecasts and derivation of TC wind-speed as hazard" of section "Methods") used for the impact forecast.

The displacement obtained with ECMWF tracks (0.5-days lead-time) shows an overestimation as compared to both the reported displacement from IDMC and the modelled displacement using IBTrACS, even though ECMWF tracks have a low-intensity bias. This is likely because the tracks from the ECMWF numerical weather predictions are simulated for longer times than is recorded by IBTrACS (see detailed discussion in the supplementary information S4). Currently no bias correction is made, but this could be addressed in future iterations of the model. As exposures, we use the global static population layer described in subsection "Population exposure" of section "Methods".

To account for the local adaptation to TC around the globe, one would ideally calibrate the vulnerability at high resolution. However, due to the rare nature of TCs, reported displacement data is only available for a few locations aggregated at country levels. To ensure sufficiently robust statistics, we group countries into ten different regions with at least 19 reported TCs and derive separate impact functions for each one of them (see Fig. S1 in the supplementary material). We use the 9 regions defined by Eberenz et al.[48], and we additionally separate Oceania into two regions: Australia and New Zealand, and the Pacific Islands. The Pacific Islands region impact functions are used for the displacement estimation in Fiji due to TC Yasa in Figs. 2–4.

We consider two optimisation options for the calibration. In both cases, we use the same third-order sigmoid-type candidate function. For the results shown in Fig. 2 where we use a single deterministic impact function, we calibrate one impact function per region which minimises the root mean square fraction (RMSF) error over all events in the region. This is designed to minimise the spread of relative error of single events. For all the uncertainty results shown in Figs. 3–5 where we use a family of functions to capture uncertainty, we instead calibrate one impact function per historical event by minimising the root mean square error. Thus, instead of one impact function per region, one obtains a bundle of impact functions per region. The details of the calibration and the resulting set of functions are shown in the supplementary information.

### Uncertainty and sensitivity

To characterise the weather forecast uncertainty alone (c.f. Fig. 2), the impact is computed separately for each of the forecast ensemble members with exposures and impact function remaining constant.

In general, however, an impact forecast depends non-linearly on the three input variables exposures, impact function and hazard. Therefore, uncertainties in each component may interact non-linearly. To capture these effects, we perform a global (as opposed to one-at-the-time) uncertainty and sensitivity analysis[28]. The uncertainty analysis gives information on the spread of the output variables (here the number of displaced people). The sensitivity analysis provides indices that subsume the sensitivity of a model output variable to the uncertainty of each input parameter[25,28].

To compute the global uncertainty distribution and sensitivity indices (c.f. Figs. 2, and 4) we use a standard quasi-Monte-Carlo numerical simulation approach[25,54]. We consider in addition to the forecast ensemble members one uncertainty parameter for the exposures and impact function each. For the exposures, we varied the total value of the population estimate uniformly between [80%, 120%]. For the uncertainty distribution of the impact functions per region, we use all the impact functions in a region that lie within the 80% confidence interval around the calibrated best estimate (the light blue lines in Fig. S2). We remark that this discrete uncertainty distribution covers a larger interval than the 80% interval of the error from the RMSF optimisation over all events in a region (c.f. supplementary material). We argue that this larger uncertainty range better represents the variability in vulnerability and is thus well-suited for impact forecasting.

Using the Sobol sampling algorithm[26], we generate a total of more than 8000 samples of the three input parameters. For each sample, we compute the impact at each exposure location, and from this the total number of displaced people. For the sensitivity analysis, the same samples are used to compute the first-order Sobol sensitivity indices as defined in Saltelli & Annoni[29] for the total number of displaced people, which characterise how much each individual input parameter contributes to the output metrics' variance.

### Reporting summary

Further information on research design is available in the Nature Portfolio Reporting Summary linked to this article.

## Data availability

The real-time updated TC tracks forecast are distributed under the Creative Commons CC-4.0-BY licence, accessible through the ECMWF website (https://www.ecmwf.int/en/forecasts/datasets/wmo-essential) and can be retrieved through the CLIMADA platform. The archived TC forecast tracks are available through the TIGGE platform (https://www.ecmwf.int/en/forecasts/datasets/wmo-essential). For the observed historical TC tracks used for impact function calibration, the tracks are distributed under the IBTrACS website (https://www.ncei.noaa.gov/products/international-best-track-archive). All tracks can be imported to the CLIMADA platform for calculating impacts. The recorded displacement data are publicly accessible via IDMC displacement database (https://www.internal-displacement.org/database/displacement-data). The exposure population data is obtained from the Gridded Population of the World (GPW v4) dataset, openly accessible from the Socioeconomic Data and Application Center (SEDAC) (https://doi.org/10.7927/H4JW8BX5).

## Code availability

All code necessary to reproduce the analysis is made available on https://github.com/manniepmkam/TC_displacement_forecast and permanently stored at https://zenodo.org/doi/10.5281/zenodo.13342825.

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

## Acknowledgements

C.M.K. acknowledges funding from the European Union's Horizon 2020 research and innovation program (grant agreement No. 820712). All authors would like to acknowledge Sylvain Ponserre, Maria Teresa Miranda Espinosa and all colleagues from the Internal Displacement Monitoring Centre (IDMC) for providing guidance and insights on the displacement data.

## Author contributions

P.M.K., C.F., C.M.K. and D.N.B. conceptualised the study. P.M.K. curated the TC forecast tracks from ECMWF to CLIMADA, conducted the formal analysis, undertook visualisation, and wrote the manuscript. P.M.K. and F.C. contributed to the historical TC displacement impact forecast analysis from 2017 to 2020. C.M.K., L.R., C.F. and D.N.B. supervised the analysis and contributed to the interpretation of the results. D.N.B. acquired funding. All authors reviewed and edited the manuscript.

## Funding

## Competing interests

The authors declare no competing interests.
