## [Peer Review File · Nature Communications]

Impact-Based Forecasting of Tropical Cyclone-Related Human Displacement to Support Anticipatory ActionREVIEWER COMMENTS

Reviewer #1 (Remarks to the Author):

Overall thoughts

Thanks for the opportunity to review this great paper. The article is extremely well written and concise with nice supporting graphics. In my view, it adds to an essential and growing body of work around impact forecasting by developing an easily accessible – and global – tool. This may ultimately prove beneficial to a number of governments and programs lacking decision-support tools currently. I commend the authors on a job well done. That said, I have a few suggestions that may strengthen the article, mainly involving some potential additional discussion on existing decision support tools, displacement datasets and validation, and the physical meaning behind some of the sensitivity analyses.

Abstract

“Anticipatory like” – I’m not sure what this means.

1. Introduction

“In our study, we forecast the number of people who are at risk of being displaced due to upcoming TC events.” Are you forecasting risk of displacement or the actual numbers displaced?

Impact forecast literature – Models are currently used to make displacement forecasts based on a combination of physical-social-traffic data. Murray-Tuite (2019) summarizes some of this, particularly on the traffic side, including the types of tools used to support emergency management decision-making in real time. The US’ National Weather Service is undergoing a paradigm shift toward “Impact-based decision support services” while evacuation models like CHIME (Morss et al. 2018; Watts et al. 2019), FLEE (Harris et al. 2021, 2023), and ICE (Davidson et al. 2018; Blanton et al. 2018) combine forecast information with evacuation decision and traffic models. All this to say, the authors should consider whether brief discussion of such work would help articulate the paper’s scope, audience, and contributions.

TC Yasa – Why is this the case study chosen to demonstrate its potential, given the relatively small number of observed displacement?

2. Results

2.1 Impact forecast for TC Yasa

Displacement data – I recommend the authors discuss the IDMC displacement data, including how they are calculated and the uncertainties associated with them. From my understanding, surveys, interviews, and cell phone tracking data used for empirical evacuation tracking can be tricky. I believe such discussion will provide further credence to the results, especially since it's crucial to the model calibration and validation.

Model validation – Is only the total displacement count used for validation, or are spatial variations accounted for? Are there issues with using a single outcome value to validate a PDF of predictions?

Forecast intensity – 200 km/hr is a category three storm on the SS, not a category 5 as the authors state. Furthermore, in my view, it's worth adding intensities (colors with tracks) and the observed best track onto Figure 2a to demonstrate that each track is associated with a wind speed / wind field prediction.

3. Discussion

“Our work shows that it is possible to provide skillful impact forecasts for tropical cyclones-related displacement from publicly available data in near-real time.” While the author’s demonstrated some potential with this modeling framework – which is valuable in its own right – I’m not sure the verification process was thorough enough to say its “skillful” since only one case study was shown and we don’t know much about the empirical dataset used. Furthermore, how useful is this data given the spread of displacement predictions?

“For the vulnerability, we consider the ensemble of individual impact functions calibrated to best represent each recorded displacement event in the used IDMC database (per region).” What is the range of observed displacement values? Are they skewed by one or two storms? And to clarify, the variability function only indirectly accounts for socioeconomics and other factors influencing evacuation, correct?

It makes sense why weather forecasts become less important to displacement outcomes closer to landfall, as forecast uncertainty decreases as you get closer to the event. However, the increase in vulnerability uncertainties (impact functions sensitivity index) upon landfall is less intuitive? What’s the physical meaning behind this? Or is vulnerability coming to the forefront simply because the meteorological uncertainty decreases, which is a widely known thing?

“There could be large lead times where the uncertainty contribution from the meteorological forecast is little but the contribution from the uncertainty of impact functions is large, and vice versa for shorter lead times.” This could be related to the fact that, when using ECMWF ensemble data, there are some meteorological cases that are easier to predict than others.

4. Data and Methods

“Here we assume a sigmoid curve functional form and calibrate a set of impact functions using the reported displacement data from the Internal Displacement Monitoring Centre (IDMC) database (accessed in September 2022).” – As noted earlier, it would be beneficial to describe this data, how it's calculated, the uncertainties associated with it, and range across previous storms.

“To ensure sufficiently robust statistics, we derive separate impact functions for 10 different regions (see figure S1 in the supplementary material).” – Did this improve the outcomes? Would you suspect that further delineations by country would improve it further? Similarly, what constitutes a “displacement event?”

Reviewer #2 (Remarks to the Author):

This study provides an estimation of displacement risk considering the uncertainties in TC track forecast, total population, and the spread of impact functions. The results indicate that the importance of TC meteorological uncertainties decreases as the expected landfall date approaches, while the significance of impact function uncertainties increases. While This point is easily understood. While its contribution is limited to the journal of Nature Communication.

Major comments:

1. Although the probability distributions of displaced people considering global uncertainty (Figure 3a) and at different lead times (Figure 4a) are demonstrated, it is unclear how these probability distribution results support anticipatory action in practice compared to using summary values.
2. The spatial distribution of risk is crucial in supporting anticipatory action. While the Author forecast the spatially explicit displacement numbers globally at a resolution of 150 arc seconds, it would be more meaningful to provide a spatially explicit map indicating hazard-sensitive, exposure-sensitive, and vulnerability-sensitive areas at a given lead time. Additionally, how does this sensitivity distribution change with different lead times?
3. As the Author mentioned, although the averaged impact cannot reflect the full range of possible outcomes, I am curious about how the average forecasted displacement changes with different lead times in Figure 4a. Furthermore, how should we interpret the nonlinear change in sensitivity shown in Figure 4b?
4. Although this study introduces the first TC-related displacement impact forecast, I believe the relevant results and findings would not change if the impact were replaced by TC-related economic damage or TC-related deaths. Since the authors aim to support anticipatory action by forecasting TC-related displacement, it is important to provide more targeted results, such as spatially explicit sensitivity and its change with different lead times, as mentioned earlier.
5. For a global study, this study lacks of comparisons with existing international researches. How reliable of the damage functions, the Author did not show well of its calibration results. For the method, this study did not consider precipitation-induced flood impact in determining human displacement. While rainfall-induced floods are major hazard in inland areas.

Reviewer #3 (Remarks to the Author):

Review for NCOMMS-23-58202, "Impact-Based Forecasting of Tropical Cyclone-Related Human Displacement to Support Anticipatory Action" by Pui Man Kam, Fabio Ciccone, Chahan M. Kropf, Lukas Riedel, Christopher Fairless, and David N. Bresch.

Recommendation: Revise

The manuscript presents a forecasting approach to TC impacts, specifically human displacement,

based on TC hazards as represented by the maximum wind speed. The approach is interesting and worthwhile and many of the current limitations are acknowledged. The authors could use further discussion on the intended use of the model – is the goal to serve as an upper bound for displacement or to help capture the actual displacement? Are the worst-case scenarios so extreme as to not be useful for planning purposes? All recorded displacement cases from 2017-2020 were assessed, but no statistics (summary or otherwise) were presented on whether the displacements all fell within the distributions, relations to averages, percentiles, etc., how those numbers evolved as the forecast times shortened – the focus was on the sensitivities. Additional comments follow.

Comments

1.1 Page 1, line 19: While this study may reveal a large range of possible outcomes, given the limitations the authors discuss and additional limitations with this approach, “the full range” is not being represented.

1.2 Page 5, Figure 2: The purple track in Figure 2(a) is not defined. The authors may also want to consider using a different color for the tracks associated with the lowest and highest displacement.

1.3 Page 5, lines 125-126: Particularly with tropical cyclones, calibrating against a small sample of past cases is going to lead to the potential for underestimates of potential hazards, especially when compounded with hazards that are not well correlated with maximum sustained winds. This limitation in the representativeness of past cases is not addressed by the authors.

1.4 Page 10, line 261 and page 12, line 304: The 1-minute maximum sustained wind speed as used in tropical cyclone tracks is determined at 10m above the surface, not 2m. Unclear if this is a typo, or a reduction factor is being applied that is not explicitly described in this manuscript.

1.5 Page 12, 307-309: The authors bring up the possibility of bias in the forecast tracks. This will also lead to a likely underestimate of intensity and associated impacts. Was a systematic analysis of the cases performed to see if the actual maximum sustained winds fell within the forecast distribution for different lead times?

Typos

2.1 Page 1, line 11: Add the word “actions” after anticipatory.

2.2 Page 1, line 12: Consider adding the word “more” before effective.

Authors Response to Reviewers' Comments

Research Article: "Impact-Based Forecasting of Tropical Cyclone-Related Human Displacement to Support Anticipatory Action." (Nature Communications; in review)

Authors: Pui Man Kam, Fabio Ciccone, Chahan M. Kropf, Lukas Riedel, Christopher Fairless, and David N. Bresch

We would like to thank the three anonymous reviewers for their valuable comments. We have made a concerted effort to adequately respond to each suggestion received from the reviewers. The comments and suggestions help in improving the quality of the manuscript.

The original comments from the referees are listed below, directly followed by our responses in blue and changes to the manuscript in blue and bold.

Reviewer #1 (Remarks to the Author):

Overall thoughts

Thanks for the opportunity to review this **great paper**. The article is extremely well written and concise with nice supporting graphics. In my view, it adds to an essential and growing body of work around impact forecasting by developing an easily accessible – and global – tool. This may ultimately prove beneficial to a number of governments and programs lacking decision-support tools currently. I commend the authors on a job well done. That said, I have a few suggestions that may strengthen the article, mainly involving some potential additional discussion on **existing decision support tools, displacement datasets and validation, and the physical meaning behind some of the sensitivity analyses**.

Thank you for the very positive assessment of our article and its presentation. Thank you also for the thorough and constructive comments. We agree that the suggestions provided can indeed strengthen the content of the manuscript. We detailed all the responses and revision below.

Abstract

"Anticipatory like" – I'm not sure what this means.

Author response: Thanks for pointing this out. This is a typo and we have revised the abstract.

Revised text (Abstract; P.1 line 10-12): While TCs pose hardships and threaten lives, their negative impacts can be reduced by anticipatory **actions** like evacuation and humanitarian aid coordination.

1. Introduction

"In our study, we forecast the number of people who are at risk of being displaced due to upcoming TC events." Are you forecasting risk of displacement or the actual numbers displaced?

Author response: This is an important semantic question which we have not stated clearly enough in our manuscript. In short, our model forecasts the number of displaced people as defined in the calibration data, which in this case study is the reported displacement data from the IDMC. The case study thus provides the 'number of displaced people as defined by the IDMC' for each forecast ensemble member. Hence, we could say that these numbers (Fig. 2b, bars) are the 'numbers displaced'. Assigning an equal probability to each

forecast ensemble member, one can then compute the 'risk of displacement', that is the average of the number of displaced over the ensemble members (Fig. 2b orange dashed line).

We emphasised this point in the results section [bold emphasise for clarity] and hope this clarifies the used language for the reader.

Revised text (Results; P.5 line 120-124):

The average forecasted number of people at **risk of displacement** in Fiji is 172, 000 (orange dashed line in figure 2b), whilst the total **number of displaced people** ranges from 3, 500 to 450, 000 for the different tracks in the forecast ensemble. The distribution of the **number of displaced people** per ensemble member, as shown in figure 2b, is right skewed with a long tail of high impacts.

Impact forecast literature – Models are currently used to make displacement forecasts based on a combination of physical-social-traffic data. Murray-Tuite (2019) summarizes some of this, particularly on the traffic side, including the types of tools used to support emergency management decision-making in real time. The US' National Weather Service is undergoing a paradigm shift toward "Impact-based decision support services" while evacuation models like CHIME (Morss et al. 2018; Watts et al. 2019), FLEE (Harris et al. 2021, 2023), and ICE (Davidson et al. 2018; Blanton et al. 2018) combine forecast information with evacuation decision and traffic models. All this to say, the authors should consider whether brief discussion of such work would help articulate the paper's scope, audience, and contributions.

Author Response: The literature mentioned indeed could complement the content of our impact forecast tool for displacement. We have added a few sentences to describe some of the existing tools/platforms in the US. At the same time, we also take the opportunity to reinstate that our work is the first globally consistent impact forecast model for displacement, that can also be used in data-scarce regions such as Fiji, where we have demonstrated in the case study.

Revised texts (Introduction; P.2-3 line 58-67):

An impact forecast moves a step forward from conventional weather forecasts to give information on how the weather will affect people. It systematically translates the weather information into risk by combining it with social variables [8]. **There have been multiple emergency evacuation decisions support tools for tropical cyclones in the US that combine weather forecast with traffic information to support evacuation plannings (e.g. Harris et al. [9], Davidson et al. [10], and Blanton et al. [11]), and establishing platforms that provides weather risk communication with integrates societal information flow (e.g. CHIME; Morss et al. [12]). Currently, there are however no tools that provide globally consistent impact information for human displacement.** Here we introduce the first prototype of a global TC-related displacement impact forecast, that could provide more comparable, standardised, and less singular information to support decision-making for anticipatory action.

TC Yasa – Why is this the case study chosen to demonstrate its potential, given the relatively small number of observed displacement?

Author response: Fiji and alongside other Island nations are the under-studied regions (as compared to countries like the US). Since we are introducing a global impact forecast model for displacement, we take the opportunity to demonstrate the model capability in providing impact forecast information in a data-scarce region. Furthermore, the island characteristic of Fiji can clearly show the hit-or-miss scenarios, which make the probability distribution of whether there will be displacement caused by TC more interesting to study, as well as the magnitude of the impact.

Revised text (Introduction; P.3 line 91-95):

Here we demonstrate the TC impact forecast for displacement by performing an analysis for the TC Yasa that hit Fiji in December 2020 and caused the displacement of 23,414 people [2,25]. **We chose Fiji as a**

demonstrating case as the Pacific Islands are an under-studied region, and while the island characteristics enable a showcase of hit-or-miss scenarios, the probability of whether there will be impacts are important when providing the forecast information.

2. Results

2.1 Impact forecast for TC Yasa

Displacement data – I recommend the authors discuss the IDMC displacement data, including how they are calculated and the uncertainties associated with them. From my understanding, surveys, interviews, and cell phone tracking data used for empirical evacuation tracking can be tricky. I believe such discussion will provide further credence to the results, especially since it's crucial to the model calibration and validation.

Author response: We thank the reviewer's comment. Indeed the IDMC displacement data is one of the crucial parts in setting up the TC displacement forecast. We have included a summary of the displacement data collection from IDMC in the data and methods section, and the discussion of the currently reported displacement data gaps and limitations in the discussion section.

Revised text (Discussion; P.10 line 210-218):

For the vulnerability, **qualitative impact data are often not available. To the best of our knowledge, IDMC provides the most comprehensive and consistent source of global displacement data, and hence we use the IDMC displacement data for impact function calibration for the global consistency. At the same time, despite many efforts from IDMC dedicated to systematically collecting and harmonising displacement data, data gaps and inconsistencies remain [32]. Some of these gaps stem from the variability in reporting, for instance, displacement can be described as "homeless" or "moved", or included in the category "directly affected" [32]. In order to capture all the plausible displacement outcomes from the TC events, we consider the ensemble of individual impact functions calibrated to best represent each recorded displacement event in the used IDMC database (per region).**

Revised text (Data and methods; P.13-14 line 351-359):

We calibrate the displacement impact functions to the reported displacement data from the IDMC database for events recorded from 2008 to 2020. **These reported numbers of displacements can vary from a few persons to millions in some high impact events [2]. IDMC systematically collects displacement data from governmental and non-governmental institutions post-disasters, and dedicated efforts to verify and harmonise data to ensure their interoperability. All the data are curated in the global database aggregated at country level [2], and the analyses are presented in the IDMC yearly Global Report on Internal Displacement (GRID) (e.g., IDMC [52]). In this study, we take the 394 events for which IDMC has reported new displacement due to TCs as the impact data for the impact function calibration.**

Model validation – Is only the total displacement count used for validation, or are spatial variations accounted for? Are there issues with using a single outcome value to validate a PDF of predictions?

Author response: Thank you for pointing this out. It is indeed not easy to fully validate an impact model. In the present case, the validation is done using the reported numbers from the IDMC which are available at the admin 0 (countries) spatial resolution. The model in itself is spatially explicit at 4 km resolution. The PDF is a description of the uncertainty of the model prediction. We show that the reported value is well within the range of the uncertainty distribution of the model's forecasts, and in general the median of the distribution is within one order of magnitude from the reported value. This is in general considered as a good agreement in risk modelling. Note that the reported IDCM data do not include a quantitative estimate of their uncertainty, but it is known that the reported values have very large variations in their accuracy due to different reporting methods and displacement definitions as discussed above

We included more in-depth discussions on the validation of the model in the supplementary section "Impact forecast validation and biases".

New section (Supplementary Information S4; P.5-7):

New section in the Supplementary Information S4: Impact forecast validation and biases

Forecast intensity – 200 km/hr is a category three storm on the SS, not a category 5 as the authors state. Furthermore, in my view, it's worth adding intensities (colors with tracks) and the observed best track onto Figure 2a to demonstrate that each track is associated with a wind speed / wind field prediction.

Author response: We thank the reviewer in pointing out the inconsistencies of the wind speed and the storm category. We have checked the observed track from IBTrACS for TC Yasa before it makes landfall in Fiji, with maximum sustained wind reaching 146 kts (75.1 m/s).

Regarding figure 2a, we find adding the colours representing the categories of the tracks obscuring the impact map. Hence we have adopted comments from another reviewer in which we highlight the tracks which predict the best and worst case scenarios that predict the lowest and highest number of total displacement, respectively. And we have added the observed best track from IBTrACS in black line.

Revised text (Results; P.5 line 109-111):

TC Yasa reached category 5 of the Saffir-Simpson wind scale, with maximum sustained winds reaching 146 kts (75.1 m/s) before making its landfall in Fiji [27].

Revised figure (Figure 2a ; P.6):

(a) The forecast-ensemble-averaged impact map of displacement by TC Yasa in Fiji as forecasted at 00:00 UTC on 15 December 2020, two days before the TC landfall. **The black line shows the observed best track of TC Yasa from IBTrACS [27]. Grey lines show the ensemble of ECMWF forecasted TC tracks, with the blue and red lines indicating the best and worst case scenarios with respect to the forecasted total number of displacements.** (b) Distribution of the forecasted potential number of displacements in Fiji for the 51 ensemble members. The vertical lines indicate the reported (black) and mean forecasted (orange) displacement.

3. Discussion

“Our work shows that it is possible to provide skillful impact forecasts for tropical cyclones-related displacement from publicly available data in near-real time.” While the author’s demonstrated some potential with this modeling framework – which is valuable in its own right – I’m not sure the verification process was thorough enough to say its “skillful” since only one case study was shown and we don’t know much about the empirical dataset used. Furthermore, how useful is this data given the spread of displacement predictions?

Author response: We agree that the term "skillful" might be misleading and we will remove it from the mentioned sentence.

The lack of detailed impact data is a general problem, both of this model approach and of the field of impact modelling in general. To our knowledge, the database by IDMC is the most comprehensive and consistent source of global displacement data, but it may show significant differences to other data, depending on the survey approach. For global consistency, we use the IDMC database for calibration and therefore emphasise that all impact numbers reported by our model should be compared on IDMC-based numbers only.

The spread of displacement predictions is precisely why we think our probabilistic model approach is more useful than a deterministic model. The spread accurately represents the statistical uncertainty in the model. Furthermore, this approach lets us identify the "contributors" of uncertainty within our model inputs and calculate the model sensitivity towards these inputs.

We updated the first paragraph of the discussion accordingly.

Revised text (Discussion; P.9 line 185-197):

Our work shows that it is possible to provide **spatially explicit impact forecasts** for tropical cyclones-related displacement from publicly available data in near-real time. We combine the tropical cyclone (TC) track forecasts from ECMWF with the population exposure and vulnerability of displacement in the open-source probabilistic risk assessment platform CLIMADA to forecast the displacement risk by TCs. We demonstrate the impact forecast for displacement with TC Yasa that hits Fiji in December 2020. **We argue that our probabilistic approach is more useful than a deterministic model. The spread of the displacement predictions reflects the statistical uncertainty in the model,** and it is important to consider the full uncertainty distribution when making decisions for anticipatory action, rather than only employing the often-used ensemble-averaged metrics. We show that the modelled impact follows a strongly skewed distribution, and thus the averaged impact cannot reflect the full range of possible outcomes. **Importantly, the tail risk of high-impact scenarios, which can be the triggers for major disasters, is inadequately represented by average measures and must be derived from a quantified uncertainty distribution.**

“For the vulnerability, we consider the ensemble of individual impact functions calibrated to best represent each recorded displacement event in the used IDMC database (per region).” What is the range of observed displacement values? Are they skewed by one or two storms? And to clarify, the variability function only indirectly accounts for socioeconomics and other factors influencing evacuation, correct?

Author response: We calibrate the impact functions using the past recorded displacement by TCs between 2008 and 2020 from the IDMC database. The range of the observed displacement per region varies from a few to hundred thousands or even millions in some regions. Indeed the data is skewed by a few high impact storms, therefore we use an ensemble of impact functions that is fitted to each individual event instead of calibrating one impact function that minimises the error from all past events. This is done so such that the set of impact functions can represent all the possible outcomes. We have included the range of the reported numbers in the data and methods section, as well as the supplementary information section S1.

For the impact functions, it has indeed indirectly accounts for other factors influencing displacement since IDMC reports displacement numbers that are attributed to TC events. Here we emphasise this point in the revised text.

Revised text (Discussion; P.11 line 267-277):

Furthermore, our model only considers displacement as a direct impact of hazards. But people can also be displaced by indirect impacts such as the loss of access to basic services (e.g., water provision) due to cascading failures of critical infrastructures [18]. **Our impact functions are calibrated using the IDMC recorded displacement attributed to TC events which include people who have been displaced due to indirect impacts. Maximum sustained wind speed is generally considered a good proxy for compound damages afflicted by TCs [33, 35]. Thus, the modelled total impact in the area affected by TC wind should reflect displacement from direct and indirect causes, but only in areas of high wind speeds. The model could further be improved by incorporating TC sub-hazard footprints like rainfall and storm surge, or explicitly model displacement triggered by housing damages, loss of livelihoods or loss of access to critical infrastructures [37]. The modelling of the impacts of compound hazards is an ongoing scientific effort (e.g. Rossi et al. [37] and Stalhandske et al. [38]).**

Revised text (Data and Methods; P. 13 line 351-353):

We calibrate the displacement impact functions to the reported displacement data from the IDMC database for events recorded from 2008 to 2020, **which the number of displacement can vary from a few persons to millions in some high impact events [2].**

Revised text (Supplementary Information S1; P.1 line 3-5):

Here we show the grouping of the TC-prone countries with similar vulnerability. In total, there are 394 displacement events all around the globe divided into the 10 regions depicted in figure S1, **with the reported number varies from a few to millions [1].**

It makes sense why weather forecasts become less important to displacement outcomes closer to landfall, as forecast uncertainty decreases as you get closer to the event. However, the increase in vulnerability uncertainties (impact functions sensitivity index) upon landfall is less intuitive? What's the physical meaning behind this? Or is vulnerability coming to the forefront simply because the meteorological uncertainty decreases, which is a widely known thing?

Author response: Thank you for your insightful comments. With our results we can explicitly quantify this intuitive assumption that weather forecast uncertainty becomes less important towards landfall. However, it was unclear whether the model was more sensitive towards the variability in hazard or vulnerability. Given that vulnerability is typically the greatest source of uncertainty in impact models, we find that the strongly varying sensitivity indices of hazard and vulnerability for different lead times are a novel result relevant for the risk modelling community.

We emphasise that it is the first order sensitivity coefficient of vulnerability increases closer to landfall, not the uncertainty. Following the formal definition by Saltelli & Annoni (2010), the sensitivity coefficient of an input parameter (here: hazard, exposure, vulnerability) describes the contribution of the variability of that input parameter towards the variability of the result (here: displacement). Therefore, increasing sensitivity towards the vulnerability with lower lead times does not necessarily mean that the overall uncertainty increases, but that variations in vulnerability tend to dominate the variations in the forecasted impact closer towards landfall.

Note that we display changes in the uncertainty of the forecasted displacement in Figure 4a. The estimated probability distribution in forecasted displacement generally narrows towards landfall, meaning that the overall uncertainty decreases. As the uncertainty in exposure and vulnerability stays constant in our model, this can be attributed to the decreasing uncertainty in the meteorological forecast.

We also recall that while overall there is the tendency for vulnerability uncertainty to become more important than hazard intensity towards landfall (figure 5b), this is not true for all individual cases. Here we include another case study in the supplementary materials of TC Harold which caused displacements in Vanuatu in April 2020 as a counter-example, which has a low sensitivity index for the hazard but large for the vulnerability at long lead time.

To make this point clearer, we have revised the manuscript as follows.

Revised text (Results; P.6 line 160-165):

We observe a general decreasing trend of the sensitivity index for the meteorological forecast, while an increasing trend for the impact function. In the meantime, the index for the change in total population remains close to zero at all forecast lead time. **We remark that the increase in sensitivity to the vulnerability uncertainty at shorter lead times cannot be equated with an increase in uncertainty from vulnerability, but only that the relative contribution to the total uncertainty from vulnerability increases, and from forecast decreases.**

Revised text (Results; P.8 line 174-183):

Figure 5 summarises the distributions of first-order sensitivity indices for all TC events. Similar to the trends identified in figure 4b, the median of the first-order sensitivity index for meteorological forecast is the largest when the forecast lead time is longer, but decreases closer to the TC landfall. The impact function's sensitivity index, in contrast, is smaller with longer lead times but increases over time. **Hence, on average over all considered events, we find that the relative contribution to the overall uncertainty from the forecast is largest at long lead times, and from vulnerability at short lead times. This overall trend does not hold for all individual cases as exemplified with TC Harold in the supplementary Figure S4 and S5 for which also at long lead times the vulnerability uncertainty dominates.**

Revised text (Discussion; P. 11 line 248-250):

Several sources of uncertainty are not represented in our model setup **which if included could add more nuances to the previous conclusions from the uncertainty and sensitivity analysis at different lead times.**

Revised text (Data and methods; P. 14-15 line 393-402):

To characterise the weather forecast uncertainty alone (c.f. figure 2), the impact is computed separately for each of the forecast ensemble members with exposures and impact function remaining constant.

In general, however, an impact forecast depends non-linearly on the three input variables: exposures, impact function, and hazard. Therefore, uncertainties in each component may interact non-linearly. To capture these effects, we perform a global (as opposed to one-at-the-time) uncertainty and sensitivity analysis [28]. The uncertainty analysis gives information on the spread of the output variables (here the number of displaced people). The sensitivity analysis provides indices that subsume the sensitivity of a model output variable to the uncertainty of each input parameter [22, 28].

New section (Supplementary Information S3; P.4-5):

New section in the Supplementary Information S3: Case study: TC Harold in Vanuatu

“There could be large lead times where the uncertainty contribution from the meteorological forecast is little but the contribution from the uncertainty of impact functions is large, and vice versa for shorter lead times.” This could be related to the fact that, when using ECMWF ensemble data, there are some meteorological cases that are easier to predict than others.

Author response: Thank you for pointing out this interlinkage. We would argue more generally saying that this is true for any meteorological model and not specific to ECMWF. We added a brief note on this in text.

Revised text (Discussion; P.11 line 242-246):

There **can** be large lead times where the uncertainty contribution from the meteorological forecast is little but the contribution from the uncertainty of impact functions is large, and vice versa for shorter lead times. **For example, this was the case with TC Harold, which impacted Vanuatu and displaced 80,000 people in April**

2020 (further details in Supplementary Information S3). Such instances could be related to the fact that some meteorological situations are easier to predict than others.

4. Data and Methods

“Here we assume a sigmoid curve functional form and calibrate a set of impact functions using the reported displacement data from the Internal Displacement Monitoring Centre (IDMC) database (accessed in September 2022).” – As noted earlier, it would be beneficial to describe this data, how it's calculated, the uncertainties associated with it, and range across previous storms.

“To ensure sufficiently robust statistics, we derive separate impact functions for 10 different regions (see figure S1 in the supplementary material).” – Did this improve the outcomes? Would you suspect that further delineations by country would improve it further?

Author response: Thank you for the excellent question, which is a quite central problem in impact modelling based on the hazard, exposures and impact function (vulnerability) framework. In short, we did not try to assess whether the regions quantitatively improve the outcomes. We expect that the uncertainties in the IDMC reported data and the ECMWF forecasts would currently dominate the signal. Furthermore, the sparsity of events (after all, cyclones are extreme events) and the relatively short IDMC records prevent us from attempting a delineation at a lower level (such as the country level). However, we expect that future improvements in the reported data and the forecast tracks will allow us to refine the impact function calibration with finer geographical delineations.

More precisely, the regions were defined based on those described in Eberenz et al. (2021) who calibrated the regional impact function for tropical cyclone impacts on physical assets using the EMDAT database. Eberenz et al. (2021) defined the regions based on 1) cyclone basin 2) minimum number of cyclones per region 3) over or underestimation of impacts using the Emanuel (2011) calibrated function. Thus, the so defined regions ensure sufficiently robust statistics by requiring a minimum number of events per region, while capturing some of the expected socio-cultural vulnerability variations. We verified that each region has a sufficient number of tracks in our work (each region has at least 19 number of events). We also assumed that the size difference between the pacific islands and Australia requires them to be split into 2 separate regions.

To make this point more clear to the reader, we have updated the sentence.

Revised text (Data and methods; P. 14 line 376-378):

To ensure sufficiently robust statistics, **we group countries into ten different regions with at least 19 reported TCs and derive separate impact functions for each one of them (see figure S1 in the supplementary material).**

Similarly, what constitutes a “displacement event?”

Author response: “Displacement” refers to when people are forced to leave their homes because of external stressors, including natural hazards such as TCs. IDMC collects displacement data when there are reports on new displacement or the movement of people resulting from disasters IDMC (2019). Here in our study, a TC displacement event refers to an event that IDMC has reported displacement which attributed the cause to TCs.

Revised text (Data and methods; P. 13-14 line 351-359):

We calibrate the displacement impact functions to the reported displacement data from the IDMC database for events recorded from 2008 to 2020, which the number of displacement can vary from a few to millions in some high impact events [2]. **IDMC systematically collects displacement data from governmental and non-governmental institutions post-disasters, and dedicated efforts to verify and harmonise data to ensure their interoperability. All the data are curated in the global database aggregated at country level [2], and the analyses are presented in the IDMC yearly Global Report on Internal Displacement (GRID) (e.g., IDMC [52]). In this study, we take the 394 events that IDMC has reported new displacement due to TCs as the impact data for the impact function calibration.**

Reviewer #2 (Remarks to the Author):

This study provides an estimation of displacement risk considering the uncertainties in TC track forecast, total population, and the spread of impact functions. The results indicate that the importance of TC meteorological uncertainties decreases as the expected landfall date approaches, while the significance of impact function uncertainties increases. While This point is easily understood. While its contribution is limited to the journal of Nature Communication.

We thank the reviewer's constructive comments. We consider our findings novel for the risk modelling community.

First, risk modellers typically focus on (uncertainties in) exposure and vulnerability, because these tend to dominate the overall uncertainty in most risk models. In our application, we can quantify explicitly that the model sensitivity towards the meteorological forecast may be equal to that towards the vulnerability, and that this relation may change drastically for different lead times. Additionally, the homogeneous uncertainty in exposure can be neglected in our modelling approach. Both results cannot be derived from the model setup alone, but must be quantified explicitly using our or similar methods.

Second, we show that the uncertainty in forecasted displacement does only decrease slightly towards landfall. Even for a relatively certain meteorological forecast with lead times of a day and less, displacement uncertainty stays high due to the uncertainty in vulnerability. This is an important result for decision makers and emergency responders, as anticipatory actions and disaster relief might then be based on knowledge about local communities and their particular needs, instead of expert advice by meteorologists.

Third, we provide a fully open-source and access globally consistent impact-based forecast model for tropical cyclones, directly applicable in data rich and data scarce region. The model furthermore seamlessly integrates the uncertainty and sensitivity assessment. Our model uses CLIMADA, which is a well-maintained Python package which would be suitable for integration into an operational setting. We believe this is a significant contribution that can find applications both for global humanitarian actors and local responders.

We revised the introduction and discussion to better contextualise our findings, stated below in the point-by-point response.

Major comments:

1. Although the probability distributions of displaced people considering global uncertainty (Figure 3a) and at different lead times (Figure 4a) are demonstrated, it is unclear how these probability distribution results support anticipatory action in practice compared to using summary values.

Author response: As visible in Figs. 3a and 4a, probability distributions of risk models are typically skewed, with a long tail towards high impact events. While probabilities for high impact events are generally low, these are the most devastating. Any risk management aims at protecting against a certain event severity, although generally the probability decreases with increasing severity. A prime example are levees designed for withstanding, e.g., river floods with return periods of 100 years. While communities are usually prepared for dealing with recurring, high-probability events, emergency responders and political decision makers focus on the low-probability events that cause major disasters. Summary values alone may inadequately convey the risks of such low-probability high-impact events. We revised the first paragraph of the discussion to emphasise these points.

Revised text (Discussion; P.9 line 190-197):

We argue that our probabilistic approach is more useful than a deterministic model. The spread of the displacement predictions reflects the statistical uncertainty in the model, and **it is important to consider the full uncertainty distribution when making decisions for anticipatory action, rather than only employing the**

often-used ensemble-averaged metrics. We show that the modelled impact follows a strongly skewed distribution, and thus the averaged impact cannot reflect the full range of possible outcomes. **Importantly, the tail risk of high-impact scenarios, which can be the triggers for major disasters, is inadequately represented by average measures and must be derived from a quantified uncertainty distribution.**

2. The spatial distribution of risk is crucial in supporting anticipatory action. While the Author forecast the spatially explicit displacement numbers globally at a resolution of 150 arc seconds, it would be more meaningful to provide a spatially explicit map indicating hazard-sensitive, exposure-sensitive, and vulnerability-sensitive areas at a given lead time. Additionally, how does this sensitivity distribution change with different lead times?

Author response: We agree that it is indeed more meaningful to have included a spatially explicit map. We have included a map plot of the largest first-order sensitivity indices at each grid point in Fiji at 2 days lead time in Figure 3c. In addition, we have plotted the ensemble averaged forecasted displacement map overlaid with the ECMWF TC forecast tracks and IBTrACS similar to figure 2a at lead time of 3 days, 1.5 days and 0 day, respectively, in figure S3 a-c, juxtaposed with the largest sensitivity indices map in Figure S3 d-f.

As seen in the sensitivity maps, larger areas in Fiji have the largest sensitivity index attributed to the hazard when the lead time is large, corresponding to the larger spread of the TC forecast tracks. When the TC is approaching landfall, the spread of the forecast tracks is smaller (more confident in the forecast location), hence most part of Fiji has the largest sensitivity index attributed to the impact function.

Revised figure (Results; P.7 Figure 3c):

(a) Probability distribution of the forecasted potential number of displaced people in Fiji due to TC Yasa with two days' lead time for each impact model run in the global (including exposure, hazard and vulnerability uncertainty) uncertainty and sensitivity analysis. **The black dashed line indicates the number of reported displacements from IDMC. The orange dashed line represents the forecasted mean from the 51 ensemble members of the TC forecast (from Figure 2a), and the purple dashed line represents the mean forecasted displacement from the global uncertainty and sensitivity analysis.** (b) The Sobol first-order sensitivity indices for the total number of displaced people, the error bars represent the 95th percentile confidence interval for each index. (c) **The largest Sobol first-order sensitivity indices at each grid point.**

New figure (Supplementary Information S2; P.3 Figure S3):

(a-c) The forecast-ensemble-averaged impact map of displacement by TC Yasa overlaid with ECMWF forecast tracks (grey lines) and the actual track from IBTrACS (black line; [4]) at 3-, 1.5-, and 0- days lead time, respectively. **(d-f)** The largest Sobol first-order sensitivity indices at each grid point, corresponding to different lead time forecast from (a-c). FM, TP, and IF denote the uncertain input parameters: forecast members, total population, and impact function, respectively.

3. As the Author mentioned, although the averaged impact cannot reflect the full range of possible outcomes, I am curious about how the average forecasted displacement changes with different lead times in Figure 4a. Furthermore, how should we interpret the nonlinear change in sensitivity shown in Figure 4b?

Author response: We agree that adding the average lines can improve the readability of the figure. We have added the lines indicating the average of the global uncertainty analysis in figure 4a. For the sensitivity analysis shown in figure 4b, because the impact forecast depends non-linearly on the interaction of the three input variables, the first order sensitivity indices simply reflect the relative contribution of each input variable. We have revised the text to make this point clearer.

Revised figure (Results; P. P.8 Figure 4):

(a) Probability distribution of the impact forecast at different forecast lead times ranging from 3.5 days to 0 days from the landfall of TC Yasa at Fiji, with the black dashed line indicating the number of reported displacement from IDMC and the red dashed line the mean forecasted displacement. (b) First-order sensitivity indices of the different uncertainty parameters for the total number of displaced people at different forecast lead times.

Revised text (Results; P.6 line 160-165):

We observe a general decreasing trend of the sensitivity index for the meteorological forecast, while an increasing trend for the impact function. In the meantime, the index for the change in total population remains close to zero at all forecast lead time. **We remark that the increase in sensitivity to the vulnerability uncertainty at shorter lead times cannot be equated with an increase in uncertainty from vulnerability, but only that the relative contribution to the total uncertainty from vulnerability increases.**

4. Although this study introduces the first TC-related displacement impact forecast, I believe the relevant results and findings would not change if the impact were replaced by TC-related economic damage or TC-related deaths. Since the authors aim to support anticipatory action by forecasting TC-related displacement, it is important to provide more targeted results, such as spatially explicit sensitivity and its change with different lead times, as mentioned earlier.

Author response: We agree that it is important to also consider other types of impact to estimate the severity of a disaster. However, in this work, we focus on the perspective of humanitarian agencies and decision makers taking anticipatory action and preparing disaster relief in the wake of a possible catastrophe. Several previous studies have shown that economic damages can be estimated with approaches similar to ours (e.g., Eberenz et al., 2021, Meiler et al., 2022). However, the spatial distribution of economic assets value and population may differ vastly, and hence the impacts will not necessarily be comparable. Also, economic damages are most important to insurers, whereas humanitarian agencies focus on human-centred impacts. Of these, displacement

is relatively clearly defined (as opposed to "affected population"), and displaced people are in dire need of shelter, clean food and water, and medical attention, which strains emergency responses. Deaths, on the other hand, often are singular, tragic events that do not necessarily provide the statistical means to calibrate a model. We updated the discussion with our reasoning for focussing on displacement, and encourage future research to incorporate estimates of other impacts in forecasting systems.

Revised text (Discussion; P.11-12 line 279-287):

We further remark that for the planning of anticipatory action, not only the displaced people, but also the people that may decide to remain in the impacted area [39], or even become trapped there if escape routes are blocked [40, 41] may be of relevance. Those trapped are often the most vulnerable, but this effect is not captured in our analysis. **Additionally, displacement is only one of the numerous socio-economic impacts related to TCs. While forecasts of asset damages and other impacts are possible with approaches similar to ours, we focus on displacement because it is arguably the most relevant impact for humanitarian agencies.** Future research is encouraged to disentangle the complex nature of disaster displacement and incorporate them in the next iterations of impact forecast systems, **and to integrate additional types of impact for a more complete picture on impending disasters.**

5. For a global study, this study lacks of comparisons with existing international researches. How reliable of the damage functions, the Author did not show well of its calibration results. For the method, this study did not consider precipitation-induced flood impact in determining human displacement. While rainfall-induced floods are major hazard in inland areas.

Author response: To the best of our knowledge, we provide the first model capable of forecasting TC-related potential displacement globally. It is therefore difficult to compare our findings to existing research, especially regarding the calibrated impact functions. We agree that the inclusion of sub-hazards of TCs, like rainfall, rainfall-induced river and pluvial floods, and storm surge, should be incorporated for a more accurate displacement model. However, this is a focus of ongoing research in the TC modelling community. Models for rainfall, storm surge, and floods are vastly more complicated than the parameterized windfield model used in our study. Previous research (e.g. Czajkowski & Done, 2014; Gettelman et al., 2017) has shown that wind speed is a good proxy for all TC-related damages. Studies like the ones by Rossi et al. (2024) indicate that it might be more worthwhile to focus on "secondary effects" of displacement, like the loss of livelihoods or access to critical infrastructure. While such additions to our model would generally be feasible, we consider them well outside the scope of the presented study. We have revised the discussion section to highlight these points.

Revised text (Discussion; P.11 line 267-277):

Furthermore, our model only considers displacement as a direct impact of hazards. But people can also be displaced by indirect impacts such as the loss of access to basic services (e.g., water provision) due to cascading failures of critical infrastructures [18]. Our impact functions are calibrated using the IDMC recorded displacement attributed to TC events which include people who have been displaced due to indirect impacts. **Maximum sustained wind speed is generally considered a good proxy for compound damages afflicted by TCs [33,35].** Thus, the modelled total impact in the area affected by TC wind should reflect displacement from direct and indirect causes, but only in areas of high wind speeds. **The model could further be improved by incorporating TC sub-hazard footprints like rainfall and storm surge, or explicitly model displacement triggered by housing damages, loss of livelihoods or loss of access to critical infrastructures [37]. The modelling of the impacts of compound hazards is an ongoing scientific effort (e.g. Rossi et al. [37] and Stalhandske et al. [38]).**

Reviewer #3 (Remarks to the Author):

Review for NCOMMS-23-58202, "Impact-Based Forecasting of Tropical Cyclone-Related Human Displacement to Support Anticipatory Action" by Pui Man Kam, Fabio Ciccone, Chahan M. Kropf, Lukas Riedel, Christopher Fairless, and David N. Bresch.

Recommendation: Revise

The manuscript presents a forecasting approach to TC impacts, specifically human displacement, based on TC hazards as represented by the maximum wind speed. The approach is **interesting and worthwhile** and many of the current limitations are acknowledged. The authors could use further discussion on the **intended use of the model** – is the goal to serve as an upper bound for displacement or to help capture the actual displacement? Are the worst-case scenarios so extreme as to not be useful for planning purposes? All recorded displacement cases from 2017-2020 were assessed, **but no statistics (summary or otherwise) were presented** on whether the displacements all fell within the distributions, relations to averages, percentiles, etc., how those numbers evolved as the forecast times shortened – the focus was on the sensitivities. Additional comments follow.

Thank you for reviewing our manuscript, your comments have indeed helped us in improving the manuscript. Our model can provide spatially explicit impact forecasts for TC-related displacement globally consistently, which is one of the first of its kind. More importantly, we emphasise that for decision making on anticipatory action we should consider the full range of uncertainty of the impact forecast outcomes. In our case, we consider the interaction of the uncertainty from the three input variables, namely the TC hazard forecast, exposure of people, and the impact functions, in the global uncertainty analysis. Our probabilistic approach enables us to identify the tail risk of high-impact scenarios, which is not accessible from a deterministic model. Furthermore, we focus on sensitivity because it is more comparable between cases, unlike the magnitude of impacts (in terms of number of displacement) that varies from hundreds to millions (see figure S6a). Therefore, we focus more in the main text on the importance of including the global uncertainty analysis to reveal the spread of the possible outcomes in the forecast, and sensitivity analysis to reveal which uncertain inputs contribute most to the overall uncertainty distribution.

At the same time, we have included a new section in the supplementary information for the impact forecast validation and biases. We show the impact forecast mean from the global uncertainty analysis for TC displacement events from 2017 to 2020 at 0.5-day lead time compared to the reported number of displacement, and discuss the biases in the same section.

Revised text (Results; P.5 line 115-117):

Figure 2a shows a **spatially explicit map** of the forecast-ensemble-averaged displacement in Fiji, overlaid with the 51 ensemble forecast TC tracks from the Integrated Forecasting System maintained by the European Centre for Medium-Range Weather Forecasts (ECMWF-IFS) [23, 24].

Revised text (Discussion; P.9 line 185-197):

Our work shows that it is possible to provide **spatially explicit impact forecasts** for tropical cyclones-related displacement from publicly available data in near-real time. We combine the tropical cyclone (TC) track forecasts from ECMWF with the population exposure and vulnerability of displacement in the open-source probabilistic risk assessment platform CLIMADA to forecast the displacement risk by TCs. We demonstrate the impact forecast for displacement with TC Yasa that hits Fiji in December 2020. **We argue that our probabilistic approach is more useful than a deterministic model. The spread of the displacement predictions reflects the statistical uncertainty in the model**, and it is important to consider the full uncertainty distribution when making decisions for anticipatory action, rather than only employing the often-used ensemble-averaged metrics. We show that the modelled impact follows a strongly skewed distribution, and thus the averaged impact cannot reflect the full range of possible outcomes. **Importantly, the tail risk of high-impact scenarios, which can be the triggers for major disasters, is inadequately represented by average measures and must be derived from a quantified uncertainty distribution.**

New section (Supplementary Information S4; P.5-7):

New section in the Supplementary Information S4: Impact forecast validation and biases

1.1 Page 1, line 19: While this study may reveal a large range of possible outcomes, given the limitations the authors discuss and additional limitations with this approach, “the full range” is not being represented.

Author response: We appreciate your feedback. We have revised the wording to better reflect the scope of our findings. We changed "the full range" to "a considerable spread" in the abstract.

Revised text (Abstract; P.1 line 17-19):

We emphasise the importance of considering the uncertainties associated with hazard, exposure, and vulnerability in a global uncertainty analysis, which reveals **a considerable spread** of possible outcomes.

1.2 Page 5, Figure 2: The purple track in Figure 2(a) is not defined. The authors may also want to consider using a different color for the tracks associated with the lowest and highest displacement.

Author response: The suggestion indeed improves the clarity of the figure. We have updated figure 2a with grey solid lines denoting the ECMWF ensemble TC forecast tracks, with the blue and red lines representing the best and worst case scenarios with respect to the forecasted total number of displacements. Also we have also included the actual track of TC Yasa from IBTrACS in black line.

Revised figure (Results; P.6 Figure 2a):

(a) The forecast-ensemble-averaged impact map of displacement by TC Yasa in Fiji as forecasted at 00:00 UTC on 15 December 2020, two days before the TC landfall. **The black line shows the observed best track of TC Yasa from IBTrACS [27]. Grey lines show the ensemble of ECMWF forecasted TC tracks, with the blue and red lines indicating the best and worst case scenarios with respect to the forecasted total number of displacements.** (b) Distribution of the forecasted potential number of displacements in Fiji for the 51 ensemble members. The vertical lines indicate the reported (black) and average forecasted (orange) displacement.

1.3 Page 5, lines 125-126: Particularly with tropical cyclones, calibrating against a small sample of past cases is going to lead to the potential for underestimates of potential hazards, especially when compounded with hazards that are not well correlated with maximum sustained winds. This limitation in the representativeness of past cases is not addressed by the authors.

Author response: We agree that the hazard footprint could be improved by incorporating compounding hazards like floods. But previous studies (e.g. Czajkowski & Done, 2014; Gettelman et al., 2017) have found that maximum sustained winds are a good proxy for TC hazards when focussing on impact modelling. Furthermore, our model implicitly incorporates displacement from TC sub-hazards like rainfall and floods, and from secondary effects like the loss of livelihoods and access to critical infrastructure. The IDMC database only lists the type of hazard that caused displacement, but not the exact process (which is often difficult to determine). We therefore

calibrate our impact functions on displacement data incorporating the compounding hazard of the TC. As long as the compounding hazard and our hazard proxy (here: maximum sustained winds) correlate spatially, we expect reasonable forecast results. We updated the discussion accordingly.

Revised text (Discussion; P.11 line 267-277):

Our impact functions are calibrated using the IDMC recorded displacement attributed to TC events which include people who have been displaced due to indirect impacts. Maximum sustained wind speed is generally considered a good proxy for compound damages afflicted by TCs [33,35]. Thus, the modelled total impact in the area affected by TC wind should reflect displacement from direct and indirect causes, but only in areas of high wind speeds. **The model could further be improved by incorporating TC sub-hazard footprints like rainfall and storm surge, or explicitly model displacement triggered by housing damages, loss of livelihoods or loss of access to critical infrastructures[37]. The modelling of the impacts of compound hazards is an ongoing scientific effort (e.g. Rossi et al. [37] and Stalhandske et al. [38]).**

1.4 Page 10, line 261 and page 12, line 304: The 1-minute maximum sustained wind speed as used in tropical cyclone tracks is determined at 10m above the surface, not 2m. Unclear if this is a typo, or a reduction factor is being applied that is not explicitly described in this manuscript.

Author response: We thank the reviewer for pointing out this typo. We have double checked the documentation of the Holland model for TC wind field, it is indeed the 1-minute maximum sustained wind speed at 10m above the surface. We have also amended the manuscript.

Revised text (Data and methods; P.12 line 310-311):

We take the maximum 1-minute sustained wind speed at **10 metres** above the ground as a proxy of the TC hazard intensity.

Revised text (Data and methods; P.14 line 361-364):

The hazard events are represented by the 1-minute maximum sustained wind speed at **10 metres** from surface derived from the corresponding historical TC tracks from the International Best Track Archive for Climate Stewardship (IBTrACS) [27], and the associated wind speed computed from the [46] model as implemented in CLIMADA [43].

1.5 Page 12, 307-309: The authors bring up the possibility of bias in the forecast tracks. This will also lead to a likely underestimate of intensity and associated impacts. Was a systematic analysis of the cases performed to see if the actual maximum sustained winds fell within the forecast distribution for different lead times?

Author response: Indeed the forecasted maximum sustained winds for TCs show a systematic bias of underestimation bias when comparing the observed IBTrACS TC data with the ECMWF ensemble forecast tracks, particularly for stronger TCs (also refer to literatures such as Chan et al., 2021 and Aijaz et al., 2019; and figure S7). However, when comparing the forecasted displacement using the ECMWF tracks at a 0.5-day lead time with the reported displacement by IDMC, there is a systematic overestimation. We find this overestimation occurs because the ECMWF simulates TC tracks for a longer duration than recorded by IBTrACS. One explanation for this discrepancy is that the ECMWF TC forecast (and numerical weather predictions in general) does not differentiate the transition from tropical cyclones to extra-tropical cyclones (Owens & Hewson, 2018), whereas IBTrACS only records tracks of tropical cyclones (Knapp et al., 2010). A detailed discussion on this issue is now included in Supplementary Information S4 - Impact Forecast Validation and Biases.

Revised text (Data and methods; P.14 line 364-371):

We note that there might exist biases in terms of the TC hazard intensity between these historical tracks and the ECMWF forecast tracks [53, 54] (c.f., Section 4.2) used for the impact forecast. **The displacement obtained with ECMWF tracks (0.5-days lead-time) shows an overestimation as compared to both the reported**

displacement from IDMC and the modelled displacement using IBTrACS, even though ECMWF tracks have a low-intensity bias. This is likely because the tracks from the ECMWF numerical weather predictions are simulated for longer times than is recorded by IBTrACS (see detailed discussion in the supplementary information S4).

New section (Supplementary Information S4; P.5-7):

New section in the Supplementary Information S4: Impact forecast validation and biases

Typos

Thank you for pointing out the typos in the manuscript. We have corrected them accordingly.

2.1 Page 1, line 11: Add the word “actions” after anticipatory.

Revised text (Abstract; P.1 line 11):

While TCs pose hardships and threaten lives, their negative impacts can be reduced by anticipatory **actions** like evacuation and humanitarian aid coordination.

2.2 Page 1, line 12: Consider adding the word “more” before effective.

Revised text (Abstract; P.1 line 12):

In addition to weather forecasts, impact forecast enables **more** effective response by providing richer information on the numbers and locations of people at risk of displacement.

Reference

- Aijaz, S., Kepert, J. D., Ye, H., Huang, Z., & Hawksford, A. (2019). Bias Correction of Tropical Cyclone Parameters in the ECMWF Ensemble Prediction System in Australia. *Monthly Weather Review*, 147(11), 4261–4285. <https://doi.org/10.1175/MWR-D-18-0377.1>
- Chan, M. H. K., Wong, W. K., & Au-Yeung, K. C. (2021). Machine learning in calibrating tropical cyclone intensity forecast of ECMWF EPS. *Meteorological Applications*, 28(6), e2041. <https://doi.org/10.1002/met.2041>
- Czajkowski, J., & Done, J. (2014). As the Wind Blows? Understanding Hurricane Damages at the Local Level through a Case Study Analysis. *Weather, Climate, and Society*, 6(2), 202–217. <https://doi.org/10.1175/WCAS-D-13-00024.1>
- Eberenz, S., Lüthi, S., & Bresch, D. N. (2021). Regional tropical cyclone impact functions for globally consistent risk assessments. *Natural Hazards and Earth System Sciences*, 21(1), 393–415. <https://doi.org/10.5194/nhess-21-393-2021>
- Emanuel, K. (2011). Global Warming Effects on U.S. Hurricane Damage. *Weather, Climate, and Society*, 3(4), 261–268. <https://doi.org/10.1175/WCAS-D-11-00007.1>
- Gettelman, A., Bresch, D. N., Chen, C. C., Truesdale, J. E., & Bacmeister, J. T. (2017). Projections of future tropical cyclone damage with a high-resolution global climate model. *Climatic Change*, 146(3–4), 575–585. <https://doi.org/10.1007/s10584-017-1902-7>
- IDMC. (2019). *Disaster Displacement—A global review, 2008-2018* (p. 54). Internal Displacement Monitoring Centre (IDMC). <https://www.internal-displacement.org/sites/default/files/publications/documents/201905-disaster-displacement-global-review-2008-2018.pdf>
- Knapp, K. R., Kruk, M. C., Levinson, D. H., Diamond, H. J., & Neumann, C. J. (2010). The International Best Track Archive for Climate Stewardship (IBTrACS): Unifying Tropical Cyclone Data. *Bulletin of the American Meteorological Society*, 91(3), 363–376. <https://doi.org/10.1175/2009BAMS2755.1>
- Meiler, S., Vogt, T., Bloemendaal, N., Ciullo, A., Lee, C.-Y., Camargo, S. J., Emanuel, K., & Bresch, D. N. (2022). Intercomparison of regional loss estimates from global synthetic tropical cyclone models. *Nature Communications*, 13(1), Article 1. <https://doi.org/10.1038/s41467-022-33918-1>

Owens, R. G., & Hewson, T. D. (2018). *ECMWF Forecast User Guide*. ECMWF. <https://www.ecmwf.int/en/e-library/81307-ecmwf-forecast-user-guide>

Rossi, L., Ponserre, S., Trasforini, E., Ottonelli, D., Campo, L., Libertino, A., Panizza, E., & Rudari, R. (2024). A new methodology for probabilistic flood displacement risk assessment: The case of Fiji and Vanuatu. *Frontiers in Climate*, 6. <https://doi.org/10.3389/fclim.2024.1345258>

Saltelli, A., & Annoni, P. (2010). How to avoid a perfunctory sensitivity analysis. *Environmental Modelling & Software*, 25(12), 1508–1517. <https://doi.org/10.1016/j.envsoft.2010.04.012>

REVIEWERS' COMMENTS

Reviewer #1 (Remarks to the Author):

While my review was quite thorough, the response was even more so. I appreciate the authors efforts to address the concerns and to synthesize a great amount of information.

Reviewer #1 (Remarks on code availability):

n/a

Reviewer #2 (Remarks to the Author):

The author has already responded to my concerns quite well. I have no further suggestions.

Reviewer #3 (Remarks to the Author):

Review for NCOMMS-23-58202, "Impact-Based Forecasting of Tropical Cyclone-Related Human Displacement to Support Anticipatory Action" by Pui Man Kam, Fabio Ciccone, Chahan M. Kropf, Lukas Riedel, Christopher Fairless, and David N. Bresch.

The manuscript present a forecasting approach to TC impacts, specifically human displacement, based on TC hazards as represented by the maximum wind speed. My precious comments on this manuscript have been addressed in the supplementary section, as well as with additional acknowledgements to the current limitations of hazards representation.

Minor comment:

-Page 5, Section 2.1 implies that the variation in maximum sustained winds has no impact on displacement in this case. An explicit statement would clarify. It does raise the question whether the variation in intensity in the ECMWF ensemble provides any useful information to the variability in displacement, or if it is essentially all coming from track spread of the forecast members.

Typos

-Page 8, line 180: "whoever" should be "however"

-Supplement, page 1, line 11: "parameter V_thresh." should be "parameter V_half."

Authors Response to Reviewers' Comments

Research Article: "Impact-Based Forecasting of Tropical Cyclone-Related Human Displacement to Support Anticipatory Action." (Nature Communications; in review)

Authors: Pui Man Kam, Fabio Ciccone, Chahan M. Kropf, Lukas Riedel, Christopher Fairless, and David N. Bresch

We would like to thank the three anonymous reviewers for the second-round review. The original comments from the referees are listed below, directly followed by our responses in blue and changes to the manuscript in blue and bold.

Reviewer #1 (Remarks to the Author):

While my review was quite thorough, the response was even more so. I appreciate the authors efforts to address the concerns and to synthesize a great amount of information.

Author response: We thank reviewer #1 for the positive feedback.

Reviewer #1 (Remarks on code availability):

n/a

Reviewer #2 (Remarks to the Author):

The author has already responded to my concerns quite well. I have no further suggestions.

Author response: We thank reviewer #2 for the positive feedback.

Reviewer #3 (Remarks to the Author):

Review for NCOMMS-23-58202, "Impact-Based Forecasting of Tropical Cyclone-Related Human Displacement to Support Anticipatory Action" by Pui Man Kam, Fabio Ciccone, Chahan M. Kropf, Lukas Riedel, Christopher Fairless, and David N. Bresch.

The manuscript present a forecasting approach to TC impacts, specifically human displacement, based on TC hazards as represented by the maximum wind speed. My precious comments on this manuscript have been addressed in the supplementary section, as well as with additional acknowledgements to the current limitations of hazards representation.

Author response: Thank you for the constructive comments. We have revised the hazard description and corrected the typos in the manuscript.

Minor comment:

-Page 5, Section 2.1 implies that the variation in maximum sustained winds has no impact on displacement in this case. An explicit statement would clarify. It does raise the question whether the variation in intensity in the ECMWF ensemble provides any useful information to the variability in displacement, or if it is essentially all coming from track spread of the forecast members.

Author response: Thank you for the comment. We agree that the text is not clear enough. We have now revised the text to improve the clarity.

Revised text (Results; P.5 line 120-123): The average forecasted number of people at risk of displacement in Fiji is 172,000 (orange dashed line in 2b), whilst the total number of displaced people ranges from 3,500 to 450,000 based on the maximum 1-minute sustained wind fields calculated from the 51 ensemble forecast

TC tracks.

Revised figure caption (Figure 2a; P.5): (a) The forecast-ensemble-averaged impact map of displacement by TC Yasa in Fiji as forecasted at 00:00 UTC on 15 December 2020, two days before the TC landfall. The black line shows the observed best track of TC Yasa from IBTrACS [27]. Grey lines show the ensemble of ECMWF forecasted TC tracks, with the blue and red lines indicating the best and worst case scenarios with respect to the forecasted total number of displacements. (b) Distribution of the forecasted potential number of displacements in Fiji **based on wind fields calculated from 51 ensemble member tracks shown in (a)**. The vertical lines indicate the reported (black) and the mean forecasted (orange)

Typos

Thank you for pointing out the typos in the manuscript. We have corrected them accordingly.

-Page 8, line 180: “whoever” should be “however”

Revised text (Results; P.8 line 180-182): This overall trend does **however** not hold for all individual cases as exemplified with TC180 Harold in the supplementary Figures S4 and S5 for which also at long lead times the sensitivity to the vulnerability uncertainty is largest.

-Supplement, page 1, line 11: “parameter V_{thresh} .” should be “parameter V_{half} .”

Revised text (Supplementary Information S1; P.1 line 9-11): where the 1-minute sustained maximum wind speed is $V(x)$, the minimum wind speed when displacement starts to occur is parameter V_{thresh} , and the wind speed at which half of the total impact occurs is parameter V_{half} .